# Predictive language comprehension in Parkinson's disease

Katharine Aveni[1], Juweiriya Ahmed[2¤a], Arielle Borovsky[3], Ken McRae[2], Mary E. Jenkins[4], Katherine Sprengel[1], J. Alexander Fraser[4,5], Joseph B. Orange[6,7], Thea Knowles[2¤b], Angela C. Roberts[1,6]*

**1** Roxelyn and Richard Pepper Department of Communication Sciences and Disorders, Northwestern University, Evanston, IL, United States of America, **2** Department of Psychology, Western University, London, ON, Canada, **3** Department of Speech, Language, and Hearing Sciences, Purdue University, West Lafayette, IN, United States of America, **4** Department of Clinical Neurological Sciences, Schulich School of Medicine and Dentistry, Western University, London, ON, Canada, **5** Department of Ophthalmology, Western University, St. Jo122seph's Health Care, London, ON, Canada, **6** School of Communication Sciences and Disorders, Western University, London, ON, Canada, **7** Canadian Centre for Activity and Aging, Western University, London, ON, Canada

¤a Current address: Temerty Faculty of Medicine, University of Toronto, Toronto, ON, Canada
¤b Current address: Department of Communicative Disorders & Sciences, University at Buffalo, Buffalo, NY, United States of America
* angela.roberts@uwo.ca

**Data Availability Statement:** R packages used in the analysis are publicly available. Data from the norming studies are publicly available at https://doi.org/10.18131/g3-6r76-cq21 and https://doi.org/10.18131/g3-aran-nz90. Participant-level norming

## Abstract

Verb and action knowledge deficits are reported in persons with Parkinson's disease (PD), even in the absence of dementia or mild cognitive impairment. However, the impact of these deficits on combinatorial semantic processing is less well understood. Following on previous verb and action knowledge findings, we tested the hypothesis that PD impairs the ability to integrate event-based thematic fit information during online sentence processing. Specifically, we anticipated persons with PD with age-typical cognitive abilities would perform more poorly than healthy controls during a visual world paradigm task requiring participants to predict a target object constrained by the thematic fit of the agent-verb combination. Twenty-four PD and 24 healthy age-matched participants completed comprehensive neuropsychological assessments. We recorded participants' eye movements as they heard predictive sentences (*The fisherman rocks the boat*) alongside target, agent-related, verb-related, and unrelated images. We tested effects of group (PD/control) on gaze using growth curve models. There were no significant differences between PD and control participants, suggesting that PD participants successfully and rapidly use combinatory thematic fit information to predict upcoming language. Baseline sentences with no predictive information (e.g., *Look at the drum*) confirmed that groups showed equivalent sentence processing and eye movement patterns. Additionally, we conducted an exploratory analysis contrasting PD and controls' performance on low-motion-content versus high-motion-content verbs. This analysis revealed fewer predictive fixations in high-motion sentences only for healthy older adults. PD participants may adapt to their disease by relying on spared, non-action-simulation-based language processing mechanisms, although this conclusion is speculative, as the analyses of high- vs. low-motion items was highly limited by the study design. These findings provide novel evidence that individuals with PD match healthy adults in their ability to

study data is available at https://doi.org/10.18131/g3-6sgw-st33, and the norming study analysis script is available at https://doi.org/10.18131/g3-ghmw-5026. Eye-tracking data and analysis scripts and deidentified demographic and neuropsychological testing data are publicly available in a collection at https://digitalhub.northwestern.edu/collections/1d4cede9-d8d6-4576-994d-91d36bd15b0b, at the following DOIs: https://doi.org/10.18131/g3-c8bg-bw89, https://doi.org/10.18131/g3-c3sr-y518, https://doi.org/10.18131/g3-tvpt-wt89, https://doi.org/10.18131/g3-41q4-6d83, https://doi.org/10.18131/g3-70sr-aq65, https://doi.org/10.18131/g3-b6fy-kb66, https://doi.org/10.18131/g3-3dy7-4g33, https://doi.org/10.18131/g3-kgcx-pw80, https://doi.org/10.18131/g3-vj5g-3t61, and https://doi.org/10.18131/g3-yjve-fs54. Picture stimuli and experiment Builder code for delivering the experiment have been archived at the Northwestern University Library and are available by request to the corresponding author.

**Funding:** This work was supported by a Parkinson's Canada Pilot Grant to authors KM, MEJ., JAF, and AR. And by gifts from an anonymous donor to MEJ and from Ms. Wendy Schall to AR. The funders had no role in study design, data collection and analysis, decision to publish, or preparation of the manuscript.

**Competing interests:** The authors have declared that no competing interests exist.

use verb meaning to predict upcoming nouns despite previous findings of verb semantic impairment in PD across a variety of tasks.

## Introduction

### Background

Among adults aged 65 and older, Parkinson's disease (PD) is the second most common neurodegenerative disorder globally [1, 2]. Characterized by a resting-state tremor, rigidity, bradykinesia, and/or postural instability, PD is often accompanied by secondary motor and non-motor features including cognitive changes that may eventually progress to dementia [3–7]. These impairments result, in large part, from the progressive loss of dopaminergic neurons in substantia nigra pars compacta [8] and by the disruption of neural connections among basal ganglia structures and diverse cortical regions [9–11]. Language impairments in PD include impaired processing of action words and concepts (e.g., [12–15]), impaired comprehension of complex syntactic structures (e.g., [16]), impaired spoken language production marked by reduced information content (specifically less complete/accurate event structures), increased frequency of grammatical errors [17–20], and difficulties interpreting figurative language and semantic ambiguities [21–23]. Because language prediction in healthy adults relies in part on complex combinatorial and event simulation mechanisms [24], language prediction is plausibly altered in people with Parkinson's and related diseases. However, it remains unclear whether PD impairs combinatorial language prediction based on agent- and verb-specific semantic knowledge.

**Verb processing impairments in PD.** Verb processing deficits are widely reported in Parkinson's disease and have been demonstrated across a variety of comprehension and production tasks. People with PD have been shown to perform worse than control participants in semantic-based verbal fluency and action fluency tasks [25, 26], even in the absence of cognitive impairment [27, 28]. People with PD show selective deficits in action word processing [12, 15, 29–32] including in verb production and naturalistic discourse tasks [13, 33–37]. Several possible accounts of verb processing deficits in PD have been provided.

It has been argued that executive function/attention impairments explain language deficits in PD [38–41]. Colman, Koerts [42] found that PD participants were impaired at producing verbs within a sentence context and that the degree of impairment significantly correlated with performance on set switching and working memory tasks. Additionally, altered lexical-semantic priming in PD [43–46] has been attributed to potential disruption of the anterior cingulate loop [43] and associated disruptions in executive function [47]. For example, Copland [43] presented polysemous words (e.g., *bank (money)-bank (river)*) to participants and found that multiple word meanings were primed for a significantly extended period of time in PD participants compared to controls, suggesting that altered attention-mediated processing may drive lexical-semantic impairments. Similarly, PD participants show poor comprehension of syntactically complex sentences and long-distance dependencies [48–52], perhaps because attention/executive control is needed to process sentences that have complex mappings between thematic (semantic) roles and syntactic structures [16, 51, 53].

Alternatively, proponents of embodied cognition suggest that semantic memory is distributed across, and grounded in, modality-specific sensory, motor, and emotion systems [54, 55]. Support for this theory includes evidence that the motor cortex, parietal cortex, and mirror neuron system are active during action-language processing (e.g., [56–61]); that action

language processing affects overt motor performance [62]; and that application of transcranial magnetic stimulation to motor brain regions may decrease response times or amplitudes of motor evoked potentials to associated verbs or sentences [63–65]. In addition, action-language networks appear to involve not only motor cortex and respective mirror neuron systems but also cortical-subcortical systems [12]. Therefore, embodied cognition theories predict that the motor impairments characteristic of PD could impair action concept and verb processing even in the absence of cognitive impairment [66]. Action and body motion verbs have been shown to be more affected by PD than non-action verbs [13, 29, 37]. Additionally, people with PD without mild cognitive impairment have shown poorer comprehension of high action content discourse passages than low action content passages [35]. Studies that examined participants' executive function abilities showed that they did not explain these action-language deficits [30, 67]. Furthermore, Roberts, Nguyen [15] showed that the degree of action word processing impairment relates to the degree of motor impairment in the action-associated limbs.

Deficits in representation and knowledge of real-world events may also explain verb processing impairments in PD, as the Two-Level Theory of verb meaning posits that event structure templates form an essential part of verb representations [68]. However, little work to date has investigated event-semantic deficits in PD. Godbout and Doyon [18] asked people with PD to produce scripts describing sequences of complex activities. They found more sequencing and intrusion errors in PD than in controls, leading the authors to suggest that changes in frontostriatal loops in PD may affect event representations. This finding is consistent with Roberts and Post [20], who found that individuals with PD generated fewer event casts (main story units grounded in an action event) than controls when producing spontaneous narratives. If event knowledge deficits are a symptom of Parkinson's disease, then cognitively intact participants with Parkinson's disease may show impaired processing of verbs and of their event-based semantic associates, compared to healthy adults. Online language processing may be particularly challenging for people with PD, considering that healthy adults activate event knowledge both to process and to predict language as it unfolds in real time.

**Language prediction in healthy adults.** In healthy adults, on-line sentence comprehension rapidly uses verb-specific syntactic and semantic information to predict upcoming words and structures [69, 70]. Healthy adults are slower to read information that violates selectional restrictions, such as a requirement for a verb's object to be animate [71, 72]. Similarly, when listening to sentences, adults saccade to a target object more quickly when the verb's selectional restrictions uniquely identify a target object than when the verb is nonselective [73]. Furthermore, healthy adults may predict upcoming language not only from syntactic constraints [74–76] but also from event-based thematic fit information—the plausibility that a given noun phrase serves as the agent (or patient, goal, etc.) of a verbal predicate [72, 77, 78]. For example, in healthy young adults, *cop* is considered a typical agent of *arrested* but not a typical patient of *arrested*; longer reading times were found for sentences that violated this thematic fit expectation (e.g., The cop arrested by the detective was guilty of taking bribes; [79]). As further evidence of semantic-based language prediction, in several ERP experiments, it has been shown that semantically constrained sentences produce a semantic prediction potential in cortical locations that reflect the semantic features of the anticipated stimuli [80–83]. Importantly, healthy adults may go beyond word-pair associations and anticipate upcoming material by combining event-based knowledge activated by an agent noun with the event-based knowledge of either a noun or a verb [84–86]. This anticipatory effect has been demonstrated even when possible targets are not constrained by visual context [87]. Thus, healthy adults spontaneously predict upcoming semantics and syntactic structures even in situations in which prediction is contingent on multiple sources of probabilistic semantic information.

Interestingly, Pickering and Garrod [88] suggest that healthy adults are able to predict others' language production because production and comprehension skills are intertwined. Inspired by motor control theories, Pickering and Garrod suggest that speakers create forward models—essentially, internal predictions of their own articulations—allowing speakers to compare their predicted utterances to their actual utterances as they unfold. Listeners may then use a perceived utterance to infer the production command and then use that (inferred) production command to simulate the speaker's language output. In support of this theory, Pickering and Garrod note that: 1) interlocutors' speech production on a shared topic often overlaps during conversational discourse, 2) language comprehension may interfere with language production [89], and 3) there appear to be shared neural pathways supporting language production and comprehension [90, 91]. Importantly, under this theory, because healthy adults predict upcoming language not only using semantic relations but also by (motor speech) simulation, language production impairments may also impair language comprehension and prediction-by-simulation. Thus, given the complexity of PD symptoms across motor and cognitive domains, even subtle impairments may significantly disrupt everyday language processing in PD.

**Language prediction in PD.** Interestingly, people with PD have exhibited deficits in non-linguistic prediction tasks that require using probabilistic, implicitly learned information [92, 93]. In a study of sensory decision-making, people with PD failed to appropriately adjust their response biases in situations where participants should have implicitly learned that one outcome was more common [94]. Similarly, in an eye-tracking study, Hochstadt [50] reported that PD participants showed an exaggerated bias towards incorrectly interpreting subject nouns as agents in passive sentences; in this study, for a subset of participants who showed overt comprehension errors, impaired online processing correlated with set-shifting deficits. Implicitly learned thematic fit information may be affected similarly in PD, particularly during online language processing. However, previous studies in PD have largely investigated offline language abilities and language priming effects and in some cases may have unknowingly included participants with cognitive impairment due to their use of brief cognition screening tools. Relatively few published studies have investigated predictive processing based on thematic fit in participants with PD without cognitive impairment. In a self-paced reading task in which participants were asked to indicate if and when a sentence "stopped making sense," Whiting, Copland [95] found that people with PD were less sensitive to violations of verb selectional restrictions than controls. This study demonstrated on-line differences in sentence processing in PD but was implemented in a self-paced reading study and relied on relatively coarse-grained manipulations of thematic roles such as animacy violations. Santerre [96] recorded eye movements using a visual world paradigm to compare healthy older adult and PD participants' abilities to predict target objects based on selective verbs (e.g., *eat*) versus non-selective verbs (e.g., *see*). Surprisingly, despite previous findings showing that PD participants experience difficulty accessing action concepts, PD participants showed a similar proportion of fixations on the target objects as did controls in the selective verb trials. The groups' similarity in anticipatory gaze patterns suggests that people with PD show intact on-line processing in relatively simple predictive contexts, in which participants are required to activate and use the meaning of only a single verb.

The experimental methods used in these existing studies may not be particularly sensitive to subtle differences in predictive processing. While there is no broad consensus on the nature of the mechanism(s) supporting language prediction, Huettig [24] suggests that a minimum of four predictive mechanisms are needed to fully account for language prediction abilities in healthy adults–a production-based mechanism, a simple associative mechanism, a combinatorial mechanism, and an event simulation-based mechanism. Under Huettig's account, even

when simple associative prediction ability is intact in PD, "smarter" combinatorial routes to language prediction may be impaired. Thus, what remains unclear is whether language prediction deficits appear in PD in situations that require rapid integration of concepts for *combinatorial* processing, as in combining agent-based and verb-based sources of semantic information.

## Present study

**Objectives and hypothesis.** Central to the present study are findings that have shown that event knowledge informs thematic fit processing [84–86]. Our objective was to test, using highly sensitive experimental methods and in a well-characterized group of participants, whether Parkinson's disease impairs online language prediction based on combinatorial thematic fit information (e.g., using the combination of *fisherman* and *rocks* to predict the target object *boat*; or using the combination of *grandmother* and *rocks* to predict the target object *cradle*.) We hypothesized that people with PD evince impairments in action and event semantic knowledge that limit their ability to combine thematic fit information from an agent noun and a verb. Therefore, we predicted that PD participants would fixate the post-verb object (patient) less than controls during the predictive window, because multiple images served as plausible target objects unless participants integrated agent and verb thematic fit information. In addition, because recent studies suggest that PD particularly affects the semantic representations of action verbs, we further predicted that participants with PD would be more impaired at processing action verbs (rated as having high motion content) than non-action verbs (rated as having low motion content).

Because individual differences in working memory, speed of processing, and executive function may influence predictive sentence processing [97, 98], special consideration was given to participants' cognitive status in the present study. Although attention, working memory, and executive function impairments have sometimes been implicated in language processing impairments in PD (e.g., [16, 43]), recent studies suggest that verb processing impairments are present even in PD participants without clear signs of cognitive impairment (e.g., [30, 67]). Therefore, the objective of the present study was to investigate combinatorial semantic language prediction abilities in a cohort without measurable and reported signs of cognitive impairment.

**Approach.** We used a visual world eye-tracking protocol to examine predictive language comprehension in participants with PD. Participants heard syntactically simple sentences while their eye movements to onscreen images were recorded. The timing and proportions of fixations on target versus distractor objects revealed how quickly the listener integrated relevant pieces of semantic information. The visual world paradigm provides several advantages compared to tasks involving manual responses and/or self-paced reading. First, eye-tracking does not require a manual limb response, which can be slowed in people with PD compared to healthy controls. Secondly, measuring eye movements towards images while sentences are spoken allows us to assess processing before an expected word is spoken, rather than measuring changes in processing that occur after the presentation of the word, as in reading protocols [24]. Finally, the structure of distractor images allows us to infer how multiple cues influenced participants' evolving predictions across a sentence.

Aim 1 examined whether the inability to integrate multiple sources of thematic role information, grounded in deficits in event knowledge, is a source of language impairment in PD. Participants completed a visual world paradigm study in which sentences were uniquely predictable only from the combination of the agent and verb (predictive sentences: e.g., *The fisherman*-agent *rocks*-verb *the <u>boat</u>*-patient). Each trial included an image of the target (e.g., *boat*) and three distractor images. One distractor was an image of an object semantically related to the

agent, but not to the verb (e.g., *net*), while another was related to the verb only (e.g., *cradle*). The third distractor was unrelated to the agent, verb, and target object (e.g., *quilt*). These sentences were designed to be canonical and syntactically simple so that they would not place high demands on participants' cognitive abilities. When typical listeners hear the agent noun, we expect them to fixate the target and agent-related images (with fewer fixations on the verb-related and unrelated images). Shortly after the onset of the verb, typical listeners are expected to fixate predominantly on the target object. If people with PD have difficulty integrating probabilistic semantic information from agents and verbs to predict an associated patient noun, then we predict that PD participants should show reduced proportions of anticipatory fixations or reduced rate of increase in proportions of anticipatory fixations on target (patient) objects following the onset of the verb.

Aim 2 assessed whether group differences in on-line sentence processing, if present, result merely from delays in lexical activation, impaired object recognition, or altered saccadic eye movements rather than from impaired combinatorial semantic processes. If on-line sentence processing impairments in PD are specifically caused by combinatorial semantic impairments, then we predict that adults with PD should not differ from healthy adults in simple sentences where the target objects are explicit, as found in previous work [96]. To address Aim 2, we presented participants with sentences in which the final word was not semantically associated with the verb or other words in the sentence ("baseline sentences," e.g., *Look at the drum)*. Scenes contained a target image (e.g., *drum*) and three semantically and phonologically unrelated items (e.g., *bathtub*, *strawberry*, *rope*). These baseline sentences differed in structure from the experimental, predictive sentences because they were not designed to be compared directly with the experimental sentences. Instead, because few previously published studies used eye tracking methodology with participants with PD, these sentences served as a control condition designed to ensure that PD participants showed intact lexical activation and recognition and intact saccadic eye movements.

## Method

All study procedures were approved by the Institutional Review Board at Northwestern University. Participants provided written consent for study participation and were compensated for their study participation. All data were collected and analyzed at Northwestern University in the senior author's research lab. Experiment delivery code and picture stimuli are archived with the Northwestern University Library system and will be provided by the corresponding author upon request. Norming data for all experimental stimuli, deidentified experimental data, and analysis scripts are publicly available using the link provided in the Data availability statement.

### Norming studies for stimuli used in the predictive visual world paradigm task

Ideal target items were expected to possess high event-based associations with both the agent and the verb in the stimulus sentences. Just as importantly, distractor items were required to be minimally associated with the agent and verb. Norming studies allowed for the identification and replacement of target and distractor pairings that were not rated as expected by healthy older adults.

**Participants.** Healthy older participants were recruited from research registries and community sources (e.g., flyers, support group outreach programs) using a convenience sampling approach. Eligible participants between the ages of 50 and 90 were required to: have at minimum a grade 10 education; have a minimum grade 10 reading ability based on the Quick

Adult Reading Inventory (QARI; [99]); and to speak English as their primary language (brief version of the Language Experience and Proficiency-Questionnaire score ≥7 speaking and understanding; [100]). Participants were excluded if they were unwilling to complete the survey in electronic format, had any past medical history of neurosurgical procedures, or had a medical history of major psychiatric or neurological illness. Participants with dementia were excluded with either the in-person (norming study 1) or telephone version (norming study 2) of the Montreal Cognitive Assessment (MoCA) using the respective dementia cut-off scores for each version (≥19/30 on the in-person MoCA or ≥18/22 on the telephone (MoCA; [101–104]).

Individuals who participated in Norming Study 1 were also invited to participate in Norming Study 2 because there was no overlap in items between the first and second norming studies and because the studies were conducted ~8 months apart. In norming study 2, participants were allowed to complete the study from their home computer to reduce participation burden. To align the normative sample with the anticipated ages in our PD cohort, we intentionally and equitably sampled from younger-old and older-old cohorts. Full participant demographics for individuals participating in the norming studies are presented in S1 Table. For norming study 1, we enrolled 14 participants in the younger-old cohort (M = 60.6 years) and 13 in the older-old cohort (M = 75.5 years), with an overall mean participant age of 67.7 years. For norming study 2, we again recruited participants into a younger-old cohort (N = 17, M = 58.4 years) and an older-old cohort (N = 13, M = 78.4 years), with an overall mean participant age of 67.1 years.

**Materials.** Test sets were based on Kamide, Altmann [85] in which sentence sets included two agents and two verbs, yielding four agent-verb combinations and thus four target objects. Following Borovsky, Elman [105], stimuli sets were designed such that each object served once as the target, once as an agent-related distractor, once as a verb-related distractor, and once as an unrelated distractor so that items were fully counterbalanced across the study:

1. The fireman rides the **truck** / bike / hamburger / candy.

2. The fireman tastes the **hamburger** / candy / truck / bike.

3. The girl rides the **bike** / truck / candy / hamburger.

4. The girl tastes the **candy** / hamburger / bike / truck.

In this example, *truck* and *hamburger* serve both as targets and as agent-based distractors for sentences (1) and (2), and as verb-related and unrelated distractors in sentences (3) and (4). As a result, if, for example, participants more strongly associate *fireman* with *truck* than with *hamburger*, then increased fixations to the target (*truck*) in sentence (1) should be offset by decreased fixations to the target (*hamburger*) in sentence (2), since each sentence was heard by each participant.

In norming study 1, we tested 32 items (8 base sentences x 4 different agent-verb combinations) reported by Borovsky, Elman [105] and augmented those with 32 additional items developed using norms from McRae et al. (https://sites.google.com/site/kenmcraelab/norms-data). We extracted all possible agent-verb, verb-target, and agent-target pairs from the sentence items. In norming study 1, each participant rated a total of 452 word pairs that were isolated from their sentence contexts (128 agent-target, 179 verb-target, 145 agent-verb). Based on its intended role in the stimulus set, each pairing was designated as either a "target" pairing (e.g., *fireman-truck*, *rides-truck*) or a "distractor" pairing (e.g., *fireman-bike*, *rides-hamburger*). Norming study 1 revealed at least one problematic component in each of the initially proposed stimuli sets for our older adult cohort. For example, the intended target pair *fireman-*

*hamburger* was rated as less commonly associated than proposed distractor pair *girl-hamburger*. Therefore, in norming study 2, additional novel test stimuli were created in a similar fashion to replace problematic stimuli/targets/distractors identified in Norming Study 1. In Norming Study 2, distractor fillers were added to the agent-verb survey, which otherwise would have contained only target pairings, resulting in a total of 744 items (281 agent-target pairs, 337 verb-target pairs, and 126 agent-verb pairs).

**Procedure.** Target objects were not necessarily required to be maximally predictable from the linguistic context because participants viewed a constrained set of four images representing the potential targets of each sentence. Target objects instead needed to be rated as significantly more probable than their on-screen distractor object counterparts. Therefore, to capture the relative strengths of association for high- and low-probability items, participants rated preselected targets using a Likert scale (rather than providing continuations in an open-ended cloze task).

Participants completed the norming study using the survey function in Qualtrics on a lab computer (Norming Study 1) or using their home computer (Norming Study 2). In Norming Study 1, participants completed the experimental blocks in a fixed order: agent-verb, agent-target, then verb-target. Questions were presented in randomized order within these blocks. However, for the agent-verb and verb-target surveys, individual items featuring the same verb were grouped together to facilitate rapid judgments. The survey questions asked participants to rate "how common" it is for various types of objects or people in the world to: "engage in various activities" (agent-verb), "participate in a single scenario" (agent-target), or to "have various actions performed on them, or performed to them" (verb-target). Participants rated the items on a scale of 1 (least common/likely) to 7 (most common/likely). Before each block, participants were given example items and ratings. Participants were instructed to provide their first impression of each object, and to use the numbers 2–6 for pairings that they believed fell between the two extremes. In norming study 2, test items were split into a form A version (338 items) and a form B version (406 items). Fifteen participants completed each form version. Study 1 revealed potential evidence of rating-fatigue, with polarized ratings for the verb-target block compared to earlier blocks, resulting in the decision to administer two form versions in the second norming study given the larger number of test items. In both norming studies, the typical time for survey completion was 45–80 minutes.

## Results

In norming study 1, target pairings were rated significantly higher (more typically related) than distractor pairings (targets' mean = 6.19, mean SD = .93; distractors' mean = 2.24, mean SD = 1.11; Welch's t-test $p < .0001$). 75.2% of individual item pairs fell within 1.5 points of the ideal value (i.e., between 5.5–7 for target item pairings and between 1–2.5 for distractor item pairings). However, all but one of the sentence test sets contained at least one component that was rated outside of these ranges. Typically, these poorly fitting items were distractors that were rated as being overly associated with agents/verbs that they were not intended to fit.

In norming study 2, target pairings were again rated significantly higher than distractor pairings (targets' mean = 5.88, mean SD = 1.09; distractors' mean = 2.29, mean SD = 1.12, Welch's t-test $p < .0001$). From the pool of all normed items (Study 1 and Study 2 combined), 12 sentence stimuli sets (48 sentences in total) were selected for the visual world paradigm experiment based on the ratings and the availability of suitable images for the target objects. Each target pair in the final set of sentence stimuli was rated higher than all corresponding distractor pairs (final targets' mean = 6.30, mean SD = .87; final distractors' mean = 1.68, mean SD = .95, Welch's t-test $p < .0001$). Individual item ratings for the final stimulus items are

publicly available at doi:10.18131/g3-6r76-cq21 [106], and motion content ratings are publicly available at doi:10.18131/g3-aran-nz90 [107].

In norming study 1, we additionally assessed whether the obtained norms were equally valid for participants of varying ages and MoCA scores. Participants' mean distances from others' ratings were 0.69 $z$ on average. Variability in item ratings did not correlate significantly with age (Pearson's $r = 0.03$, $p = 0.877$), suggesting that the norms are equally valid for younger-old and older-old participants. With the exclusion of one outlier participant, there was also no significant correlation between variability in item ratings and MoCA scores (Pearson's $r = -0.31$, $p = .127$). The lack of an effect of MoCA scores suggests that the norms are equally valid in participants with fully intact cognition as in participants with scores borderline for mild cognitive impairment (MCI).

## Motion content norming study

Once the 48 stimulus sentences were selected, a third norming study was conducted to quantify the degree of motion content of each selected verb. Participants for the motion content norming study consisted of a convenience sample of 20 healthy adults who primarily spoke English and were 18–42 years old (mean = 24.10, SD = 5.66). For this study, also conducted via Qualtrics, participants were instructed to "rate how much movement would typically be used to complete each of the following actions" using a 1–7 Likert scale. When rating items' motion content, participants asked to consider the indicated verb sense (e.g., rock [cause to move back and forth or side to side]). Mean ratings for each verb were used to classify items as 'low' (<4) or 'high' (>4) motion content in subsequent motion content eye tracking analyses, resulting in 13/48 sentences being designated as 'high' motion content.

## Experiment: Anticipatory eye movements during sentence comprehension

We assessed PD and control participants' prediction of patients (target objects) from the combination of an agent and a verb. Performance in this task was contrasted with performance on baseline sentences (*Look at the [drum]*), which did not require use of combinatory semantics.

**Participants.** Participants were recruited using methods identical to those in norming study 1. In addition to the eligibility requirements for the norming studies, participants were required to have an in-person MoCA score of 24 or higher (reflecting an optimised cut-off score for individuals with PD; [108]). Participants also were required to have sufficient vision to read the instructions and view the images used in the study on the display monitor ($\geq$ 20/50 vision, either corrected or uncorrected), not to have cataracts, and were required to have a pure tone audiometric average $\leq$ 40dB HL bilaterally at 500, 1000, and 2000 Hz, reflecting normal or only mildly impaired hearing. All PD participants had been diagnosed for a minimum of one year and were under the care of a movement disorders neurologist. Except for one *de novo* participant, all PD participants were on stable PD medication. Additionally, participants were excluded during the screening stage if the experimenter could not obtain reliable calibration for eye-tracking.

In total, 50/80 screened participants met all criteria and were enrolled. Two participants (one control and one with PD) were excluded after study completion but before data analysis due to poor-quality eye-tracking data. Poor-quality data was operationally defined as having >15% of trials removed due to track loss, coupled with a relative lack of fixations on target objects (reflecting lack of engagement with the task). The final analysis included 24 control participants (16 female, 8 male) and 24 participants with PD (11 female, 13 male). Table 1 includes demographic data for both the PD and control groups. There were no significant differences between groups on age, years of education, audiometric pure tone average, MoCA, or

geriatric depression scale scores. The Unified Parkinson's Disease Rating Scale [109] and Hoehn and Yahr [110] scores suggest that the PD cohort comprised mainly participants with relatively mild disease.

**General study procedures.** In most cases, participants completed the study in two visits, within a 7- to 10-day window. When participants had current neuropsychological assessment data available from our lab (within 6 months of the eye-tracking experiment) we reduced study burden and preserved assessment fidelity by using those existing assessment data (N = 10 PD; 1 control). In these cases, participants completed only a single study visit. Participants with PD completed the experiment in "on" medication state. While day to day fluctuations are common in PD, testing at the same time of day ensured participants were medicated at similar levels during both their testing sessions. Study visit 1 typically included the screening and neuropsychological assessment, which lasted approximately 2 hours. Study visit 2 typically included the eye-tracking tasks, which lasted approximately 1 hour.

**Neuropsychological assessment.** A detailed description of each test and the domains assessed is presented in S2 Table. Tests were administered by a single examiner and were administered according to protocols outlined in published procedural manuals. Assessment sessions were audio/video recorded for double-scoring and fidelity procedures. Disagreements in test scoring were resolved through consensus procedures. Neuropsychological test data are presented in Table 1. The absence of dementia was confirmed neuropsychologically. All participants were independent in activities of daily living, and with none scoring below 2 SD on more than one cognitive test, no participant met standard criteria for dementia [111]. Three control participants and one PD participant scored between 1.5 and 2 SD below normal limits on multiple tests and thus met criteria for mild cognitive impairment [112]. However, the majority of these participants scored below normal limits on two tests in two different domains, which may be less indicative of progressive cognitive impairment compared to failing two tests within the same domain [113]. In the present study, only one control participant was impaired on multiple tests within a single domain. In addition, these participants met the minimum cognitive screening requirement and showed essentially normal task performance upon visual inspection of the eye-tracking data. Thus, we judged that these participants' cognitive impairment was mild enough not to significantly interfere with participants' ability to complete the task, and so they were not excluded from analysis. Descriptive and inferential statistics for the neuropsychological tests were calculated using the *mean()*, *sd()*, and *t.test()* functions in R version 3.6.0.

**General eye-tracking method.** Eye movements were recorded monocularly using a desktop-mounted SR Research Eyelink 1000+ camera set to record at a sampling rate of 1000 Hz. Participants were seated with their eyes approximately 57 cm from the monitor and 52 cm from the camera with their heads stabilized using a SR Research chinrest. The computer was set to its native resolution of 1920 x 1080 and the use portion of the screen was adjusted to 30.5 cm horizontally and 25.4 cm vertically, resulting in 29.9 (width) × 25.1 (height) degrees of visual angle to accommodate the participant-monitor distance. The remaining portion of the screen was blacked out both during the experimental trials and calibration to keep luminosity consistent and to minimize inaccurate calibration due to changes in pupil size. Calibration and validation area proportions were adjusted accordingly. Experimental stimuli were presented using Experiment Builder, Version 2.1.140 software. Calibration and validation procedures were performed before each experimental block and whenever the participant moved their head and calibration became inaccurate (detected automatically at the beginning of each trial). Eyelids sometimes obscure the pupil, particularly in older adults [114], and this can lead to track loss when pupils are tracked via the centroid method [115]. Therefore, pupils were tracked via the ellipse method for all participants.

**Table 1. Participant demographic and neuropsychological data by group.**

| Variable | Mean (SD), 95% Confidence Interval [Lower, Upper] Range | | t-tests |
|---|---|---|---|
| | *PD* | *Control* | **t-tests** |
| *Demographic Data* | | | |
| Age (yrs.) | **68.02** (8.86) [64.28, 71.76] 52–89 | **66.54** (10.85) [61.96, 71.13] 53–87 | $t(46)$ = -0.52, $p$ = 0.608 |
| Education (yrs.) | **17.38** (2.73) [16.66, 18.96] 12–26 | **17.81** (2.72) [16.66, 18.96] 12–23 | $t(46)$ = 0.56, $p$ = 0.581 |
| PTA (better ear) | **21.01** (9.15) [17.15, 24.88] 8.3–35 | **17.89** (6.17) [15.28, 20.49] 5–40 | $t(46)$ = -1.39, $p$ = 0.172 |
| MoCA (/30) | **27.29** (1.63) [26.60, 27.98] 24–30 | **27.08** (1.67) [26.38, 27.79] 24–30 | $t(46)$ = -0.44, $p$ = 0.663 |
| GDS (/15) | **1.79** (2.50) [0.74, 2.85] 0–10 | **0.75** (0.90) [0.37, 1.13] 0–3 | $t(28.82)^d$ = -1.92, $p$ = 0.065 |
| Handedness (self-report) | 91.7% Right, 8.3% Left | 87.5% Right, 12.5% Left | - |
| MDS-UPDRS-III (/132)[a] | **23.7** (11.4) [18.90, 28.56] 6–44 | - | - |
| Hoehn & Yahr (/5)[b] | **1.77** (0.75) [1.45, 2.09] 1–4 | - | - |
| Years since PD diagnosis | **5.99** (5.49) [3.67, 8.31] .9–25 | - | - |
| *Neuropsychological Assessment Data* | | | |
| D-KEFS Trail Making Test (Switching vs. Number/Letter scaled score) | **9.58** (2.98) [8.33, 10.84] 3–17 | **9.04** (2.10) [8.16, 9.93] 6–13 | $t(46)$ = -0.73, $p$ = 0.470 |
| D-KEFS Color Word Interference Test (Switching vs. Color + Word scaled score) | **10.92** (2.00) [10.07, 11.76] 6–14 | **11.58** (2.41) [10.56, 12.60] 7–16 | $t(46)$ = 1.04, $p$ = 0.303 |
| Digit Span Backwards (scaled score) | **8.78** (2.89) [7.56, 10.00] 3–16 | **8.79** (3.37) [7.37, 10.22] 3–15 | $t(46)$ = 0.01, $p$ = 0.992 |
| Semantic Fluency (Animals, scaled score) | **11.75** (3.14) [10.42, 13.08] 6–19 | **11.00** (2.27) [10.04, 11.96] 7–16 | $t(46)$ = -0.95, $p$ = 0.347 |
| CLOX I (Free-draw scaled score) | **10.96** (1.81) [10.20, 11.72] 6–13 | **11.42** (1.18) [10.92, 11.91] 9–13 | $t(39.55)^d$ = 1.04, $p$ = 0.304 |
| CLOX II (Copy trial scaled score) | **9.17** (2.87) [7.96, 10.38] 1–12 | **9.38** (3.46) [7.91, 10.84] 3–12 | $t(46)$ = 0.23, $p$ = 0.821 |
| Boston Naming Test (/30) | **27.67** (2.24) [26.72, 28.61] 22–30 | **28.13** (2.33) [27.14, 29.11] 22–30 | $t(46)$ = 0.70, $p$ = 0.491 |
| Northwestern Naming Battery (/31) | **30.74** (0.53) [30.52, 30.97] 29–31 | **30.75** (0.68) [30.46, 31.03] 28–31 | $t(46)$ = 0.02, $p$ = 0.984 |
| NAVS sentence comprehension (/30) | **29.88** (0.34) [29.73, 30.02] 29–30 | **29.75** (0.53) [29.53, 29.97] 28–30 | $t(38.97)^d$ = -0.97, $p$ = 0.337 |
| Pyramids and Palm Trees (/14) | **13.92** (0.28) [13.80, 14.04] 13–14 | **13.96** (0.20) [13.87, 14.04] 13–14 | $t(46)$ = 0.59, p = 0.561 |
| HVLT-R total recall (t-score) | **47.38** (9.16) [43.51, 51.24] 30–66 | **49.67** (7.19) [46.63, 52.70] 32–64 | $t(46)$ = 0.96, $p$ = 0.340 |
| BVMT-R total recall (t-score)[e] | **52.83** (9.06) [49.01, 56.66] 34–68 | **53.96** (14.85) [47.69, 60.23] 20–73 | $t(38.1)^d$ = 0.32, $p$ = 0.753 |

*Note*. PTA = Pure Tone Average (of 500, 1000, and 2000 Hz). MoCA = Montreal Cognitive Assessment. GDS = Geriatric Depression Scale. D-KEFS = Delis-Kaplan Executive Function System. NAVS = Northwestern Assessment of Verbs and Sentences. HVLT-R = Hopkins Verbal Learning Test- Revised. BVMT-R = Brief Visuospatial Memory Test-Revised.

[a]All PD participants were assessed using the Unified Parkinson's Disease Rating Scale motor section (MDS-UPDRS-III; [109]) and the modified Hoehn and Yahr Scale [110] by a certified examiner. The UPDRS-III is a standardized measure of motor impairment in PD and assesses tremor, slowness (bradykinesia), stiffness (rigidity), and balance.

[b]The Hoehn and Yahr Scale is a standardized measure of disease severity, with scores ranging from 0 (asymptomatic) to 5 (wheelchair-bound).

[c] Calculated by averaging participants' scaled scores from: digit span backwards, D-KEFS trail-making test switching versus composite number and letter sequencing, D-KEFS color-word inhibition test switching versus composite color and word reading, semantic fluency, and CLOX I [112]. Scaled scores for the semantic fluency and CLOX I tests were converted from t-scores and z-scores, respectively.

[d]lower d.f. due to Welch's test for unequal variances

[e]BVMT-R normative values for participants aged 80–89 were drawn from Gale, Baxter, Connor, Herring, and Comer (2007)

We used color images of each of the target objects in both predictive and non-predictive trials to enhance recognizability [116]. All sentences were recorded in an ANSI-standard audiometric testing booth by a native Midwestern American English male speaker. Recordings were made in 44.1 kHz sampling rate and mono format using an AKG C 520 head-mounted condenser microphone connected to a laptop via a SoundDevices USBPre 2 pre-amplifier. See S1 and S2 Appendices for additional detail on image selection, audio recording, audio stimuli selection, and editing procedures.

Each eye-tracking session began with a comfortable listening level task to ensure stimulus audibility given the increased risk of hearing impairment in older adults. Participants heard a practice list of words, and the experimenter raised or lowered the volume progressively, introducing new lists of phonologically similar practice words, until a volume level was reached where 100% of words were repeated accurately and/or the participant indicated that the volume could no longer be increased without becoming uncomfortably loud.

**Visual world paradigm procedures.** To address Aim 1, participants listened to predictive sentences in a visual world paradigm task. In the predictive sentence condition, the agent was more associated with the target object and an agent-based distractor object than with the verb-based and unrelated distractors. The verb was more associated with the target object and a verb-based distractor object than with the agent-based and unrelated distractors.

However, interpretation of group differences in anticipatory looks to target objects can be accounted for by factors unrelated to predictive processes, such as systematic group differences in saccadic latencies or systematic issues with the prerequisite steps of quickly identifying pictured objects, maintaining objects' locations, and rapidly comprehending simple spoken sentences. Therefore, we also included a baseline sentence condition modeled on Santerre [96] that included sentences in the form *Look at the [target]*. Null group effects on the baseline condition would argue against these alternative interpretations, allowing more robust conclusions regarding the role of language prediction on the predictive sentence condition. The two conditions were presented in a fixed order with the baseline sentence condition following the predictive sentences for all participants.

**Visual world paradigm stimuli.** A schematic of the eye tracking task from the participant's perspective is shown in Fig 1 below. In order to moderate visual bias that would potentially obscure predictive eye movements, we allowed a 2000 millisecond preview period.

Each stimulus set contained four sentences, with each image serving once as the target object, once as an agent-related distractor, once as a verb-related distractor, and once as an unrelated distractor. Each participant saw the same stimuli. However, the order of sentences was pseudorandomized into ten different pre-determined trial orders using a random number generator. When necessary, items were rearranged manually to prevent the target object from appearing in the same quadrant of the screen on more than two consecutive trials and to prevent sentences from the same set appearing consecutively (e.g., *The fisherman **rocks** the boat* could not be followed by *The grandmother **rocks** the cradle*). There were 8 practice trials and 48 experimental trials of this sentence type (S3 and S4 Tables; high motion content sentences marked 'H').

Baseline sentence stimuli were presented in a single block at the end of the eye tracking session, following all predictive sentences, so that the differing task directions would not influence participants' performance on predictive sentences. Baseline sentences followed the same randomization procedures as the predictive trials. During each of these trials, participants viewed four objects that were phonologically and semantically unrelated to each other and that were not present in the prediction trials. Items were balanced so that each object appeared three or four times. In addition, each trial contained at least one object that was never used as the target object. This was done to minimize participants' ability to predict the target object by ignoring

## A: Predictive Sentences

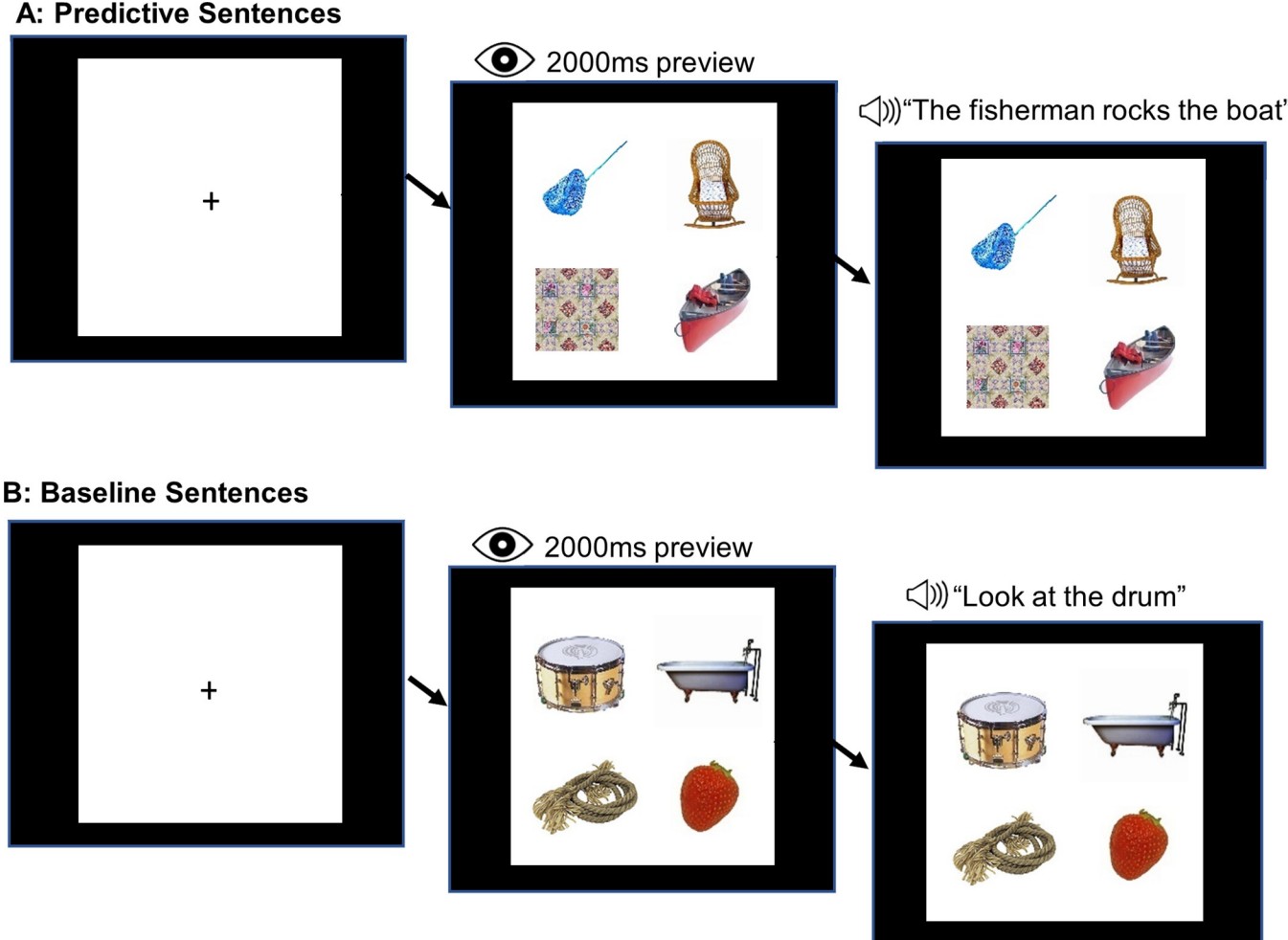

## B: Baseline Sentences

**Fig 1.** A schematic of the trial sequence, (A) for predictive sentences and (B) for baseline sentences. The left-most panel depicts the fixation cross, which participants had to fixate in order to launch the trial. The middle panel depicts the preview period when participants simultaneously previewed the four items. The right-most panel shows the screen layout when the sentence audio began. Images remained onscreen for the full duration of the sentence. Images were displayed in a balanced quadrant in the center of a white background, with ~1.5 inches between images horizontally, ~0.5 inches between images vertically, and ~1 inch between the edge of the image and the edge of the white portion of the screen. Stimuli images displayed in Fig 1 are similar but not identical to the original images and are therefore for illustrative purposes only. Pool net and strawberry reprinted from Brodeur, Dionne-Dostie [117] under a CC BY license, with permission from Mathieu Brodeur, original copyright 2009, 2010. Cradle [match_0188] and bathtub [object_0075] reprinted from Kovalenko, Chaumon [118] under a CC BY license, with permission from Niko Busch, original copyright 2012. Rope reprinted from Brady, Konkle [119] stimulus set; canoe reprinted from Konkle and Caramazza [120]; drum reprinted from Konkle and Oliva [121] under a CC BY license, with permission from Talia Konkle, original copyright 2008, 2011, 2013. Quilt original photo under a CC BY license, with permission from H.R. Templeton C.D. Hancok, original copyright 2021.

objects that had been the target of previous sentences. There were 5 practice trials and 20 experimental trials featuring baseline sentence stimuli (S5 and S6 Tables).

To ensure that participants were attending to the sentences, attention check questions were added at randomly chosen intervals throughout the predictive trials (2 questions during practice items, 12 in experimental items). Questions were manually re-arranged as needed to appear at least two sentences apart. During these comprehension question trials; participants saw a single word on-screen along with the prompt: *Say "YES" if you heard this word in the last sentence*: *[word]*. *Otherwise, do not say anything*. Words that had not appeared in the most recent trial also did not appear anywhere else in the experimental sentences, making it relatively easy for participants to identify the unused words. Participants answered all questions

correctly except for one participant, who responded to two of the questions by repeating the onscreen word but otherwise answered all questions correctly. There were no such questions for the non-predictive sentences.

Instructions for the predictive sentence blocks were presented simultaneously verbally and onscreen as follows: "*You will hear a sentence and see four images on the screen. Pay attention to the sentence and the pictures. You may look around the images freely. Sometimes we will ask you whether you heard a certain word in the last trial. When this happens, if you just heard that word, say yes. If you did not just hear that word, please wait quietly.*" Instructions for the baseline sentences were presented verbally and onscreen as follows: "*You will see a display with four pictures while hearing a sentence. Look at whatever picture you are told to look at. We will not ask you any questions.*"

**Pre-processing of eye-tracking data.** Data were exported using the sample report in Eye-Link Data Viewer software package (SR Research Ltd., version 3.2.48) and thus included both longer fixations and shorter fixations broken up by saccades. The eye-tracking data were pre-processed using a custom script in R (version 3.6.0) that made extensive use of the *eyetrackingR* package [version 0.1.8; 122]. Fixations outside of the four interest areas were excluded from analysis. The *eyetrackingR* package was used to subset the data into agent-only, verb-only, and target-only analysis windows for the predictive trials and a target-only window for the non-predictive trials. To account for saccade programming time, all time windows were created with a 200ms delay from the auditory onset of the relevant word [123, 124]. The data were binned into 50ms intervals in order to mitigate eye-movement based dependencies [125]. Time windows were extended or truncated slightly as needed to avoid creating bins with systematically fewer samples. For the predictive trials (Aim 1), the beginning of the agent window was extended 24ms into the first article, the end of the verb window was extended 37ms into the second article, and the end of the target object window was truncated by 4ms. For the baseline trials (Aim 2), the end of the target object window was extended by 25ms.

*EyetrackingR* functions were also used to assess and to remove trials with 25% or more track loss during the full sentence duration [126] and to calculate proportions of fixations on each interest area for each 50ms time bin in the analysis windows. The mean number of predictive trials remaining was 47.25 ± 1.66 SD. The mean number of baseline trials remaining was 18.65 ± 2.85 SD. In 27.7% of predictive trials and 25.1% of baseline trials, participants were already fixating the target image just prior to sentence onset. However, removing these trials would have resulted in substantial data loss and would result in artificial increases in fixations to the target object early on due to mere regression to the mean [127, 128]. Therefore, in our statistical analysis, we included trials that began with fixations to the target image. In S7–S10 Tables, we repeated these same analyses with target-anticipated trials excluded. This resulted in only minimal changes to the model fit and no changes to the pattern of findings between PD and control participants.

**Analysis of eye-tracking data.** The eye-tracking observations used across all study aims were nested within participants and therefore did not meet the independent samples assumption for ANOVAs. Instead, the cleaned, subsetted eye-tracking data were fitted to a series of logistic mixed effects (multi-level) models using the glmer command in the R package *lme4* version 1.1–21 [129]. All intermediate proportions of fixations on each interested area were rounded to either 0 (no fixation) or 1 (fixation) because the raw binned proportions essentially followed a binomial distribution. Therefore, rather than predicting these proportions, our models predict the odds ratio of fixations on the target versus fixations on all other distractors. This odds ratio is log-transformed into "logits" of fixations on each interest area. In addition, each clustering unit (e.g., participant) is permitted to have its own intercept, and in many cases, its own slope term as well. The fixed effects estimates are drawn from the average of

these individual intercepts and slopes. The random effects characterize the degree of difference across individual intercepts and slopes.

For each time window of interest, we tested for significant differences in proportions of fixations on the target object over time, by group, and for group by time interactions (which represent differential effects of group on rate of increase in fixations on the target). To avoid collinearity issues in our assessment of time effects, we used the orthogonal time polynomials generated by the *eyetrackingR* package instead of natural time polynomials [130]. As a result, intercept values reflect mean values rather than values at t = 0. Linear term estimates indicate whether fixation proportions increase, decrease, or remain flat between the beginning and end of the analysis window. Finally, the quadratic term estimates indicate whether fixation proportions change at a constant rate or a changing rate. In each model, we additionally generated random intercepts and random linear slopes for subjects (to assess individual differences) and for items (to assess stimulus-driven variability). Random quadratic slopes for subjects and items were included only when they did not cause singular model fits (overfitting). Random effects were estimated with an unstructured covariance matrix. Mixed effects models of binary data have been reported previously (e.g., [131–133]). In addition, simulation research suggests that growth curve analyses of binary data are feasible and result in accurate parameter estimates when sample sizes are sufficiently large, e.g., >200 for linear estimates and >1000 for quadratic estimates [134], as was the case in the present study (48 participants * 47 trials ≈ 2256 trials per model, with 10–14 measurement occasions per trial).

For predictive sentences, we modeled fixations on the target object in separate logistic models for three time windows of interest: the agent window, the verb window, and finally the target window. We additionally modeled the effect of group on fixations on the agent- and verb-related distractor objects. We then performed additional analyses designed to explore the role of the motion content of the stimuli and of PD participants' motor abilities on fixations to the target object. Specifically, we modeled the effect of motion content (and its interactions with group) in the agent, verb, and target time windows. For the non-predictive sentences, we modeled only fixations on the target during the target window, following the spoken instruction "Look at the ___." Group and motion content factors were both sum-coded (control -0.5, PD +0.5; low-motion -0.5, high-motion +0.5).

## Results and discussion

### Predictive sentences

Overall graphical results for proportions of fixations in the predictive sentences are presented in Fig 2 (logits of fixations are presented in S1 Fig), followed by statistical results for each analysis window.

**Fixations on the target object.** See Table 2 for statistical results and Fig 3 for graphical illustrations of model fits. In each analysis window (agent, verb, and target), the linear term was significant and positive, indicating that fixations on the target object increased throughout the predictive sentences. The significant linear increase in proportions of fixations on the target object during the verb window suggests successful *prediction* of the target object, although target fixations did not peak until the target window. Significant negative curvature in proportion of fixations on the target was observed only for the target window, indicating that the proportion of fixations on the target objects levelled off at the end of the trials. In contrast, the proportion of fixations on the target object rose steadily throughout the verb window.

There were no significant group differences in the intercept (mean), linear, or quadratic term in any analysis window. Thus, PD and control participants' proportion of fixations and rate of increase in fixations on the target object were essentially the same throughout the

## Looks to target vs. distractor images in predictive sentences

**Fig 2. Gaze data by group in predictive sentences.** Binned binomial gaze probabilities to each area of interest, averaged across subjects and trials over the duration of the predictive sentences for controls (left) and PD participants (right). The x-axis reflects the elapsed time in the trial in milliseconds, offset by 200ms to account for saccade programming/launching time. The solid vertical lines mark the onset of the agent, verb, and target words. The dotted vertical line marks the end of the verb statistical analysis window (which extends only partially into the post-verb article). Error bars represent standard error of the mean.

predictive sentences, including during the critical verb window, the point at which the target object was uniquely predictable by combining agent and verb thematic fit information. Because each image was presented four times but used only once as the target object, by the end of the study, fixations on the target object upon sentence onsets may not have been entirely "naïve," and thus these trials might be thought to bias the results. However, the pattern of findings did not substantially change when these analyses were re-conducted with target-anticipated trials removed rather than included. All main effects and interactions involving

**Table 2. Analyses of PD versus control gaze logits to the target object during predictive sentences.**

| Fixed Effects | Predictive Sentences: Fixations on the Target Object | | | | | | | | |
|---|---|---|---|---|---|---|---|---|---|
| | **Agent Time Window** | | | **Verb Window** | | | **Target Window** | | |
| | *Estimate* | *S.E.* | *p* value | *Estimate* | *S.E.* | *p* value | *Estimate* | *S.E.* | *p* value |
| Intercept | -0.862 | 0.08 | **< .001** | -0.361 | 0.11 | **< .01** | 0.668 | 0.17 | **< .001** |
| Linear time | 0.473 | 0.13 | **< .001** | 0.532 | 0.10 | **< .001** | 1.055 | 0.23 | **< .001** |
| Quadratic time | 0.092 | 0.05 | 0.055 | 0.083 | 0.05 | 0.069 | -0.198 | 0.07 | **< .01** |
| Group (Control/PD) | 0.033 | 0.10 | 0.745 | 0.204 | 0.16 | 0.198 | 0.295 | 0.30 | 0.330 |
| Group x Linear | 0.037 | 0.15 | 0.807 | -0.095 | 0.13 | 0.458 | 0.336 | 0.35 | 0.343 |
| Group x Quadratic | 0.046 | 0.10 | 0.632 | -0.056 | 0.09 | 0.538 | -0.008 | 0.11 | 0.944 |

Note: Bolded values are significant at the $p < .05$ level

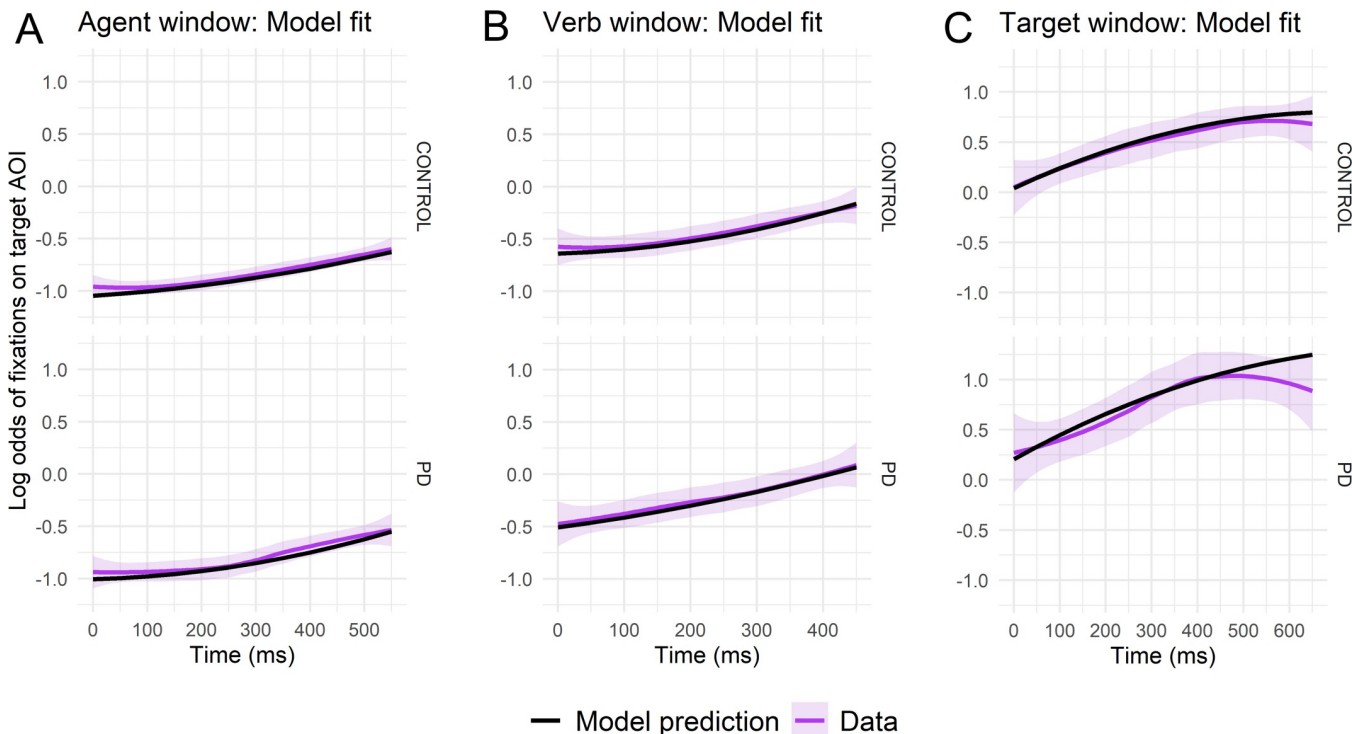

**Fig 3. Agent window model fit in predictive sentences.** Actual vs. model-predicted log odds (logits) of fixations to the target interest area in (A) the agent window, (B) the verb window, and (C) the target window, averaged across subjects and trials over the duration of each window for controls (top) and PD participants (bottom). The x-axis reflects the elapsed time in each window in milliseconds, offset by 200ms to account for saccade programming/launching time.

group remained non-significant. Significance statistics related to the overall shape of the curve changed in the following minor ways: 1) for the agent window, the quadratic term became significant ($p < .001$) due to the artificially low starting position of fixations on the target object, and 2) for the target window, the quadratic term became significant at the $p < .001$ level, rather than at the $p < .01$ level.

**Fixations on the agent-related distractor.** During the predictive window, the agent-related distractor was the critical distractor object, as continued fixations on this object would suggest that participants failed to integrate the constraining semantic information given by the verb. Therefore, as a complement to the target analyses, we also analyzed fixations to the agent-related distractor during the verb window. Overall, participants disengaged with the agent-related distractor during the verb window, as shown by a significant negative linear estimate (*Estimate* = -0.592, *SE* = 0.12, $p < .001$). The quadratic term was also significant and negative (*Estimate* = -0.275, *SE* = 0.05, $p < .001$), indicating that looks to the agent-related distractor declined more and more rapidly over the course of this time window. As with looks to the target distractor, there were no group differences in the intercept (*Estimate* = -0.178, *SE* = 0.11, $p = 0.12$), linear (*Estimate* = -0.188, *SE* = 0.17, $p = 0.27$), or quadratic terms (*Estimate* = -0.083, *SE* = 0.10, $p = 0.40$). In addition, a Welch's t-test revealed that the means of participants' fixations on the target (42.2% ± 12.1%) and agent-related distractor (32.0% ± 7.6%) differed significantly during the verb window ($t(78.957) = 4.9318$, $p < .001$), confirming that participants preferentially fixated the target image during the prediction window.

**Fixations on the verb-related distractor.** We conducted post-hoc analyses to assess the significance of the apparent "bump" in fixations on the verb-related distractor towards the end

of the predictive sentences. The agent, verb, and target windows described above were chosen based on expected patterns in gaze towards the target object specifically. Thus, they were not optimal for analyzing fixations on the verb-related distractor, which peaked approximately in the time region in between the verb and target windows. We therefore modeled a new combined time window spanning the onset of the verb through the offset of the target when analyzing the proportion of fixations on the verb-related distractor.

There were significant negative effects of quadratic time (*Estimate* = -1.142, *SE* = 0.25, $p < 0.001$) and of linear time (*Estimate* = -1.024, *SE* = 0.34, $p = < 0.01$) on fixations on the verb-related distractor. This pattern suggests that fixations on the verb-related distractor rose briefly but significantly before declining further. In addition, Student's t-test revealed that the means of participants' fixations on the verb-related distractor (13.2% ± 5.5%) and unrelated distractor (10.8% ± 5.0%) differed significantly during the verb window ($t(94) = 2.23$, $p = .028$), confirming that participants fixated the verb-related image more than the unrelated image during the post-verb window. Yet there were no significant group differences on the intercept (*Estimate* = -0.141, *SE* = 0.21, $p = 0.49$), linear term (*Estimate* = -0.361, *SE* = 0.55, $p = 0.51$), or quadratic term (*Estimate* = -0.462, *SE* = 0.35, $p = 0.18$). Therefore, as with proportion of fixations on the target object, PD and control participants did not differ in their patterns of fixations on the verb-related distractor object.

**Effect of motion content and motor abilities on fixations on the target.** Using the motion content ratings obtained from healthy younger adults, we modeled the effect of motion content and its interactions with group in the agent, verb, and target time windows (see Table 3 for statistical results). Findings relating to the shape of the curve did not change. There were significant linear increases in looks to the target across all time windows and significant negative quadratic terms for the agent and target time windows, with no significant main effects of group (PD vs. control). There was also no main effect of motion content on either the intercept, linear, or quadratic term. However, we found significant group x motion content interactions in the agent and target time windows, as illustrated in Fig 4. Specifically, for PD participants but not for control participants, the mean level of fixations on the target object during the agent window (intercept term) was higher during low motion sentences. Additionally, the linear increase in looks to the target object was less steep in high-motion sentences for

**Table 3. Effect of motion content on gaze logits to the target entity during predictive sentences.**

| Fixed Effects | Predictive Sentences: Fixations on the Target Object | | | | | | | | |
|---|---|---|---|---|---|---|---|---|
| | Agent Time Window | | | Verb Window | | | Target Window | | |
| | *Estimate* | *S.E.* | *p* value | *Estimate* | *S.E.* | *p* value | *Estimate* | *S.E.* | *p* value |
| Intercept | -0.872 | 0.09 | **< .001** | -0.372 | 0.12 | **< .01** | 0.626 | 0.17 | **< .001** |
| Linear time | 0.434 | 0.15 | **< .01** | 0.463 | 0.11 | **< .001** | 1.003 | 0.25 | **< .001** |
| Quadratic time | 0.125 | 0.06 | **< .05** | 0.042 | 0.05 | 0.428 | -0.226 | 0.08 | **< .01** |
| Group (Control/PD Intercept) | -0.029 | 0.10 | 0.774 | 0.216 | 0.16 | 0.173 | 0.277 | 0.30 | 0.362 |
| Motion Content (Intercept) | -0.055 | 0.16 | 0.726 | -0.061 | 0.19 | 0.744 | -0.174 | 0.17 | 0.294 |
| Group x Linear | 0.084 | 0.16 | 0.604 | 0.167 | 0.14 | 0.223 | 0.474 | 0.36 | 0.186 |
| Group x Quadratic | 0.031 | 0.11 | 0.780 | -0.124 | 0.11 | 0.241 | 0.084 | 0.13 | 0.517 |
| Motion x Linear | -0.183 | 0.27 | 0.497 | -0.270 | 0.21 | 0.205 | -0.196 | 0.36 | 0.583 |
| Motion x Quadratic | 0.126 | 0.11 | 0.256 | -0.143 | 0.11 | 0.177 | -0.086 | 0.15 | 0.560 |
| Group x Motion (Intercept) | -0.257 | 0.06 | **< .001** | 0.012 | 0.07 | 0.853 | -0.088 | 0.06 | 0.151 |
| Group x Motion x Linear | 0.179 | 0.22 | 0.420 | 0.228 | 0.21 | 0.283 | 0.608 | 0.23 | **< .01** |
| Group x Motion x Quadratic | -0.112 | 0.22 | 0.612 | -0.282 | 0.21 | 0.183 | 0.331 | 0.23 | 0.146 |

Note: Bolded values are significant at the $p < .05$ level

## Looks to target vs. distractor images in predictive sentences

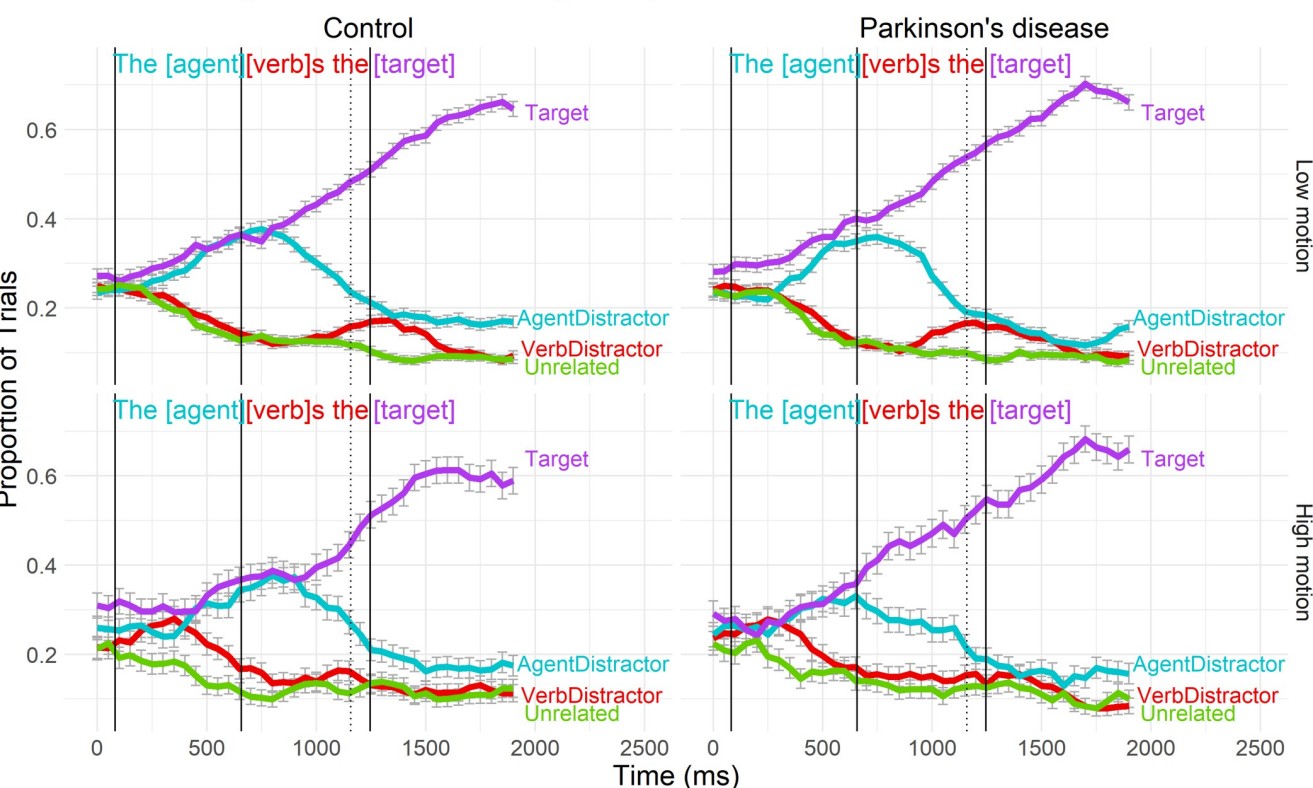

**Fig 4. Gaze data by group and motion content in predictive sentences.** Binned binomial gaze probabilities to each area of interest, averaged across subjects and trials over the duration of the predictive sentences for: Controls x Low motion (upper left), Controls x High motion (lower left), PD x Low motion (upper right), PD x High motion (lower right). The x-axis reflects the elapsed time in the trial in milliseconds, offset by 200ms to account for saccade programming/launching time. The solid vertical lines mark the onset of the agent, verb, and target words. The dotted vertical line marks the end of the verb statistical analysis window (which extends only partially into the post-verb article). Error bars represent standard error of the mean.

control participants, whereas PD participants' looks to the target object increased at a similar rate across high- and low-motion sentences.

**Discussion–predictive sentences.** In light of prior evidence of action and event knowledge impairments in PD, we predicted that PD participants would show impaired on-line sentence processing when required to combine information from an agent and verb to predict the post-verb object. Specifically, we predicted that PD participants would show reduced proportions and rates of increase of fixations on target (patient) objects. However, these predictions were not clearly supported by the results. When all trials were considered, proportions of anticipatory fixations on the target object did not differ in either mean level or rate of increase between PD and control participants. Additionally, within the PD group, motor severity did not predict fixations on the target object during either the verb or target object time window, as might be expected if sentence comprehension impairments in PD are driven largely by degradations in motoric brain regions and action language processing.

The absence of group differences in fixations on the target object during the agent window suggests that both PD and control participants successfully used agent-based (noun) information to predict the two related, likely target objects. This finding is in line with several other studies that have found intact noun processing in PD participants without cognitive impairment [12, 29–32, 135]. This finding is also in keeping with Santerre [96], who found intact predictive processing in PD in simple predictive situations.

The verb analysis window was the critical interest period, as we expected group differences to appear when participants had to rapidly integrate multiple sources of semantic information about the agent and the verb to determine which object was the target. If the PD participants had been unable to make use of the semantic information provided by the verb, then their proportion of fixations on the target during the verb window would not have risen above their proportion of fixations at the end of the agent window. Thus, if PD participants were unable to use verb semantic information, they would have shown lower proportions of fixations and/or less steep increases in proportions of fixations on the target during the verb window than controls. Instead, the PD participants showed a similar overall level of fixations on the target as controls, as well as similar rates of increase in fixations on the target, suggesting an intact ability to combine agent and verb thematic fit information to predict the target. Additionally, the lack of group differences in the target window suggests that the PD participants were as capable as controls of fixating explicitly named objects.

Because fixations outside of the interest areas were excluded from our analysis, fixations away from the target object were necessarily accompanied by increased proportions of fixations on one or more distractor images. However, fixations on the verb-related distractor object also did not differentiate PD and control groups. Furthermore, the peak in average fixations on the verb-related distractor occurred in the target window for both PD and control participants. Instead of reflecting errored predictive processing, fixations on the verb-related distractor may reflect a delayed "contrast effect" reported previously in healthy adults, in which participants double-check the current target word against a plausible contrasting object [136, 137]. Alternatively, these looks to the verb-related distractor may indicate a local, non-predictive thematic priming effect also reported previously in healthy adults [98]. In the present study, to ensure diversity of agents in each study block, each block included two sentences that had the same verb but different agents. As a side effect of this design choice, towards the end of each block, the verb-related distractor images would have been the targets of prior sentences and possibly highly primed in both participant groups as a plausible alternate target.

The results comparing participants' processing of high- and low-motion content trials did not match our initial predictions. We predicted that PD participants might be selectively impaired at processing high-motion sentences and would therefore make fewer predictive fixations to the target image in the verb window of the high-motion sentences. Instead, during the end 'target' window of the high-motion sentences, healthy older adults showed a pattern of slower increases in fixations on the target objects, whereas PD participants' fixations were unaffected by the sentences' motion content. This observed insensitivity to items' motion content may reflect an unexpected consequence of or adaptation to Parkinson's disease. It has been suggested that healthy older adults are 'less embodied' than younger adults, having been shown to favor visual processing over bodily factors in a variety of experimental tasks [138]. Similarly, Pickering and Garrod [88] propose that there are multiple routes available for language prediction, one simple associative and one motor speech simulation route. According to the 'disrupted motor grounding hypothesis,' embodied language processing mechanisms are disturbed in frontostriatal movement disorders such as PD [139]. Thus, it is possible that our participants with PD may have exhibited a form of accelerated aging or even strategic processing that decreased attention to verbs' motion content and increased PD participants' reliance on spared associative language processing mechanisms that do not rely on action simulation. In contrast, control participants appeared sensitive to the degree of sentences' motion content, even at the cost of delayed or reduced attention to the target items, perhaps because attending to motion content is not overly detrimental to everyday language processing for healthy adults. However, we interpret these results cautiously, considering that this experiment was not designed to compare high- and low-motion content trials. In particular, the significant motion

content x group interaction in the agent window is problematic as it implies that participants were sensitive to verbs' motion content *before* hearing onset of the verb. This result may have been driven by uncontrolled differences in other psycholinguistic properties of the high-motion sentences, experimental 'noise' due to the small number of these sentences, or by differences in the order of presentation of these sentences across participants.

### Baseline sentences

Graphical results for proportions of fixations in the baseline sentences, designed as a control condition to assess group differences in eye movements or lexical activation, are presented in Fig 5 (logits of fixations are presented in S2 Fig).

**Fixations on the target window.** As depicted in Fig 6, the log odds of fixations on the target object increased linearly over the course of the target window ($Estimate = 3.406$, $SE = 0.27$, $p < 0.001$). The quadratic term was also significantly positive, suggesting a slight positive acceleration of looks to the target during this window ($Estimate = 0.384$, $SE = 0.14$, $p < 0.01$). There were no significant differences in PD compared to control participants on average proportion of fixations on the target ($Estimate = 0.053$, $SE = 0.14$, $p = 0.71$), on the linear rate of increase in fixations on the target object ($Estimate = 0.223$, $SE = 0.33$, $p = 0.50$), or on the curvature of proportion of fixations on the target object ($Estimate = -0.128$, $SE = 0.23$, $p = 0.57$).

**Discussion–baseline sentences.** The lack of a main effect of group indicates that, as predicted, PD participants and healthy older adults did not differ significantly in their average

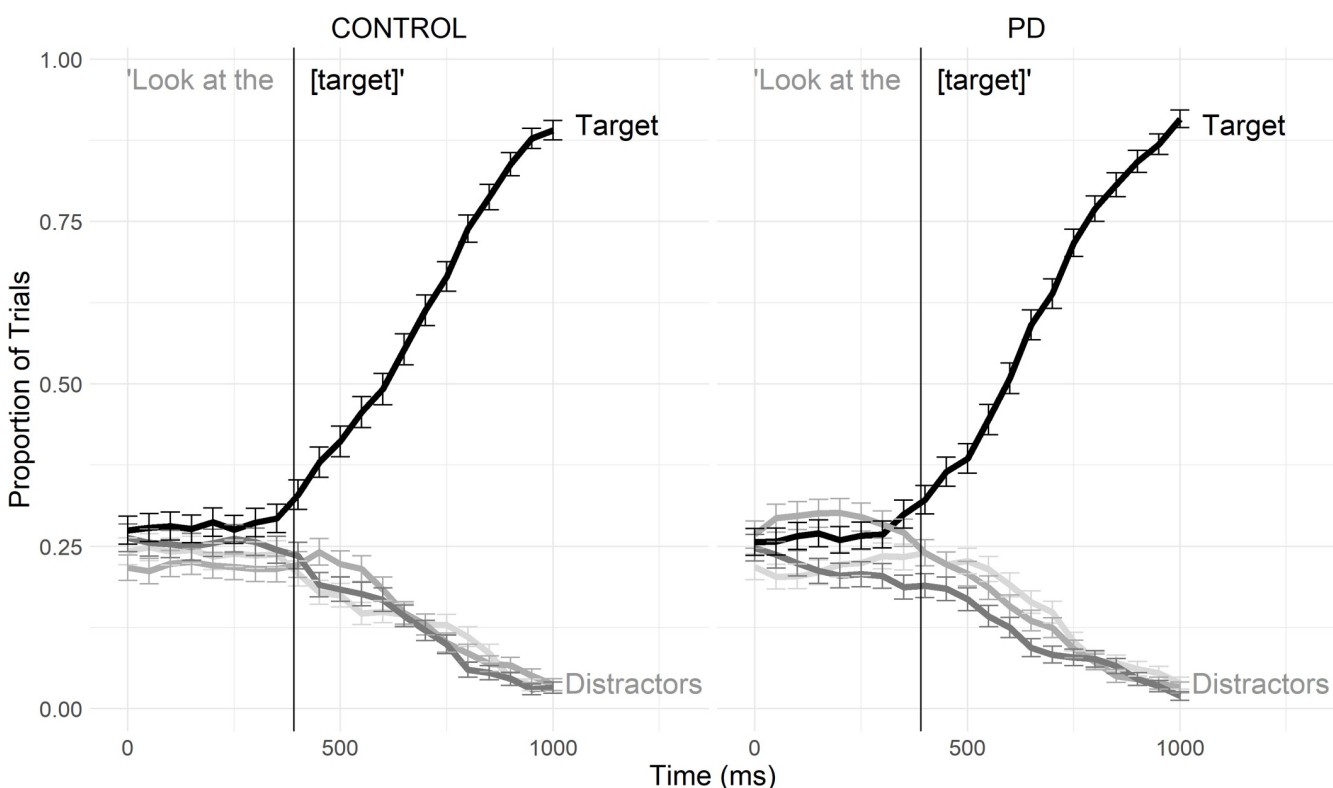

**Fig 5. Gaze data by group in baseline sentences.** Binned binomial gaze probabilities to each area of interest (AOI), averaged across subjects and trials over the duration of the non-predictive sentences for control participants (left) and PD participants (right). The x-axis reflects the elapsed time in the trial in milliseconds, offset by 200ms to account for saccade programming/launching time. The vertical line marks the onset of the target word. Error bars represent standard error of the mean.

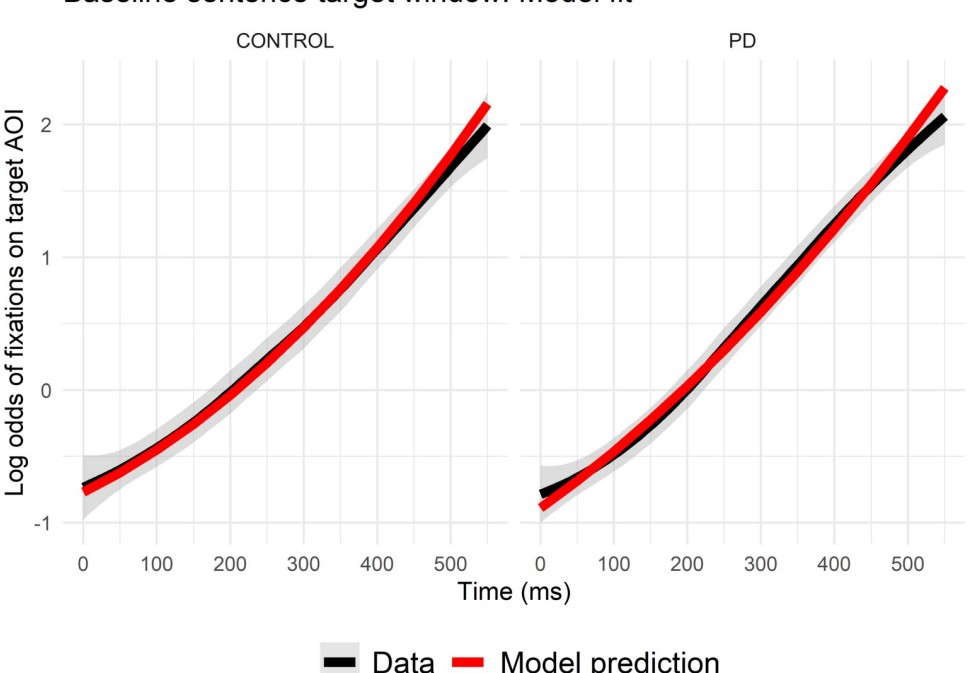

**Fig 6. Target window model fit in baseline sentences.** Actual vs. model-predicted log odds (logits) of fixations to the target interest area, averaged across subjects and trials over the duration of the target window for controls (left) and PD participants (right). The x-axis reflects the elapsed time in the target window in milliseconds, offset by 200ms to account for saccade programming/launching time.

fixations on the target objects while listening to the baseline sentences. Similarly, the lack of any significant group by time interactions shows that the groups had similar rates of increase and acceleration in proportion of fixations on the target, indicating similar processing patterns in these baseline sentences. These findings are consistent with the results from the target window in the experimental (predictive) sentences, where both groups showed equivalent proportions of fixations on target objects when the targets are named explicitly. They are also consistent with past studies reporting that people with PD demonstrate intact processing of syntactically simple sentences that do not require fine-grained on-line processing of semantic goodness of fit [49, 52]. The similarity of the obtained PD data to that of healthy adults helps validate the use of eye-tracking to study language comprehension in PD, as it suggests that saccadic behaviors in non-predictive contexts are similar between controls and individuals with PD without cognitive impairment, at least when the areas of interest are approximately 5cm$^2$ or greater.

## Conclusions

### Strengths and limitations

One strength of the present study is that it featured largely early to mid-stage PD participants who were well characterised and with neuropsychologically-determined absence of dementia. Whereas other studies investigating core action language deficits in PD participants without cognitive impairment have relied largely on cognition screening tools [29–31, 35], in the present study our stringent criteria increase our confidence in participants' cognitive status.

The experimental design of the present study also provided several advantages. First, stimuli were counterbalanced such that each object and corresponding image served once in each role (target, agent-distractor, verb-distractor, and unrelated). As a result, each item acted as its own distractor across the course of the study, minimizing the experimental impact of any items that may have been less anticipated or less easily processed in a particular sentence context. This eliminated the typical need to control the frequencies and semantic relatedness between differing sets of target and distractor items.

In addition, the fact that the sentences that participants heard in the present study were designed to be semantically plausible is a strength of the study because readers appear to routinely monitor sentences' plausibility [140] and because the presence of anomalous sentences in a protocol may disrupt normal processing of low-probability yet plausible items [141]. Furthermore, many of the distractor objects in the current study fit the sentences reasonably well, albeit at a lower probability than the target object. This design feature meant that participants were often required to detect subtle differences in real-world likelihood rather than attending only to selectional restrictions. Together, these aspects of the study design reinforce our confidence that the findings tested participants' use of fine-grained, probabilistic event knowledge. Finally, this study design allowed us to analyze the degree and timing of fixations on distractor images in addition to target images. We replicated previous findings of temporarily increased fixations on the verb distractor during the verb window [85, 98] and further showed that this effect did not significantly differ by group.

One important limitation of the present study is that we did not carefully control the action content of the verbs used in the predictive sentences. Due to norming and counterbalancing constraints, sentences, this study did not contain exclusively high-motion content verbs, for which PD participants without cognitive impairment have been hypothesized to show greater impairment (e.g., [13, 29, 37, 66]). The analysis comparing low- and high-motion sentences addresses this concern, but it is complicated by the fact that 1) there were relatively few high-motion sentences, resulting in noisy data and potentially spurious findings 2) the ordering of high- vs. low-motion sentences was not controlled during psuedorandomization, and 3) the psycholinguistic properties of high- and low-motion sentence types may have differed on average. In addition, the motion content ratings used for these analyses were completed by a relatively small number of participants.

Sentences were syntactically simple to facilitate thematic role assignment and were designed to be predictable primarily upon the application of real-world event knowledge; however, the present study design did not disentangle the precise influences of event knowledge deficits versus deficits in online verb semantic or thematic role processing. Also, in spite of the counterbalancing approach taken in the present study, the interpretation of these findings is potentially limited by some characteristics of the stimuli. For example, some items fit less well in a count use compared to a non-count use (e.g., *saves the money*). In addition, processing speed for individual items may have been affected based on the use of some verbs that were homonyms with nouns [142] and by the use of persons and scenes for some post-verb objects [143, 144]. The fact that norming studies participants rated verbal associations rather than the picture stimuli represents an additional limitation. Modelling random effects of trials helped control for the effects of individual items but may not entirely capture the effects of these features of the stimuli on online processing.

## Future directions

One important direction for future research is to assess language function in PD in a greater range of tasks. Action or event-based language deficits may emerge more prominently in

production tasks, or in relatively low-context situations such as confrontation naming or single-word comprehension. Although we did not see group differences in verb confrontation naming in the present study, this is likely attributable to the fact that our participants scored essentially at ceiling on the Northwestern Naming Battery used for the present study. Indeed, studies using other verb naming tests have repeatedly found PD participants to be impaired, particularly for action verbs [13, 15, 29, 30, 37, 135]. Thus, future work should test PD participants' language processing abilities in multiple language contexts. Future research should also test whether the finding of PD participants' insensitivity to verbs motion content relative to control participants replicates under more stringent experimental control, and whether this finding may reflect a disruption of embodied language prediction mechanisms in PD and a subsequent shift to reliance on relatively intact, non-embodied mechanisms.

## Summary

The current study sought to determine whether on-line processing of thematic fit, grounded in combinatorial event semantic knowledge, is a source of language impairment in PD. The results did not support our prediction that people with PD without cognitive impairment would differ from controls in combining semantic information to predict target objects, and instead revealed robust on-line prediction effects even in the context of disease that is argued to affect action and event knowledge. In addition, PD participants were not more impaired on high-motion than low-motion sentences, although control participants in our study did show this pattern of reduced fixations on the target objects during high-motion sentences. These findings do not appear to support strong embodied cognition theories, although they may be compatible with theories that predict that brain activity in motor regions will be reduced when processing is supported by context [145, 146] and with theories that predict disrupted use of embodied language processing mechanisms in Parkinson's disease [139]. The present study also contributes additional evidence to the existing literature showing that healthy older adults immediately use semantic information to comprehend sentences.

## Supporting information

**S1 Table. Norming studies participant demographics.**
(PDF)

**S2 Table. Neuropsychological battery.**
(PDF)

**S3 Table. Predictive sentences practice.**
(PDF)

**S4 Table. Predictive sentences, full trials (Sets 1–12).**
(PDF)

**S5 Table. Baseline sentence practice.**
(PDF)

**S6 Table. Baseline trials.**
(PDF)

**S7 Table. Analyses of PD versus control gaze logits to the target object during predictive sentences, excluding anticipated trials.**
(PDF)

**S8 Table. Analyses of PD versus control gaze logits to the agent-related object during the predictive window, excluding anticipated trials.**
(PDF)

**S9 Table. Analyses of PD versus control gaze logits to the verb-related object during the combined verb + target window, excluding anticipated trials.**
(PDF)

**S10 Table. Effect of motion content on gaze logits to the target entity during predictive sentences, excluding anticipated trials.**
(PDF)

**S1 Fig. Looks to target versus distractor images in predictive sentences, in logits.**
(PDF)

**S2 Fig. Looks to target versus distractor images in baseline sentences, in logits.**
(PDF)

**S1 Appendix. Visual stimuli selection and editing.**
(PDF)

**S2 Appendix. Word onsets in sentence audio.**
(PDF)

## Acknowledgments

The authors are grateful to Dr. Jeff Elman for his inspiration and his contributions to the conceptual development of this study, the interpretation of our pilot data, and stimuli selection for the final study. Research reported in this publication was supported in part by the NIDCD of National Institutes of Health to author AR (approximately 25% of the reported project resources). The content is solely the responsibility of the authors and does not necessarily represent the official views of the National Institutes of Health. We also thank the following individuals: Hui Zhang, Leah J. Welty, Siyuan Dong, and Matthew Walenski (statistical methods consultations); Steve Uliana (stimuli audio recordings); Marissa Esparza and Stephanie Gutierrez (data collection); Chelsea Avery, Richard Richter, Stephanie Gutierrez, Richard Martin, and Ashana Torani (stimuli selection and development); Tara Poikey, Jayna Patel, Alexander Havens, Mackenzie Barber, Erin Blaze, Madeleine Chow, Sarah Lawson, and Melanie Davis (double scoring neuropsychological assessments).

## Author Contributions

**Conceptualization:** Katharine Aveni, Juweiriya Ahmed, Arielle Borovsky, Ken McRae, Angela C. Roberts.

**Data curation:** Katharine Aveni.

**Formal analysis:** Katharine Aveni, Angela C. Roberts.

**Funding acquisition:** Ken McRae, Mary E. Jenkins, J. Alexander Fraser, Angela C. Roberts.

**Investigation:** Katharine Aveni, Katherine Sprengel, Angela C. Roberts.

**Methodology:** Katharine Aveni, Juweiriya Ahmed, Arielle Borovsky, Ken McRae, Katherine Sprengel, J. Alexander Fraser, Joseph B. Orange, Thea Knowles, Angela C. Roberts.

**Project administration:** Angela C. Roberts.

**Resources:** Juweiriya Ahmed, Arielle Borovsky, Ken McRae, Mary E. Jenkins, Katherine Sprengel, Angela C. Roberts.

**Software:** Katharine Aveni, Juweiriya Ahmed, Thea Knowles.

**Supervision:** Arielle Borovsky, Ken McRae, Angela C. Roberts.

**Visualization:** Katharine Aveni.

**Writing – original draft:** Katharine Aveni.

**Writing – review & editing:** Juweiriya Ahmed, Arielle Borovsky, Ken McRae, Mary E. Jenkins, J. Alexander Fraser, Joseph B. Orange, Thea Knowles, Angela C. Roberts.

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
