## [Decision Letter · Decision Letter 0]

2 Aug 2021

PONE-D-21-22350

Predictive language comprehension in Parkinson’s disease

PLOS ONE

Dear Dr. Roberts,

Thank you for submitting your manuscript to PLOS ONE. I have now received reviews from 2 experts in the field and read your manuscript myself. We feel that it has merit but does not fully meet PLOS ONE’s publication criteria as it currently stands. Both reviewers recommend additional analyses and clarification or elaboration on certain points. Their comments are quite clear and constructive, so I won’t reiterate them. Therefore, we invite you to submit a revised version of the manuscript that addresses the points raised during the review process.

We look forward to receiving your revised manuscript.

Kind regards,

Daniel Mirman

Academic Editor

PLOS ONE

Journal Requirements:

4. We note that Figure 1 in your submission contain copyrighted images. All PLOS content is published under the Creative Commons Attribution License (CC BY 4.0), which means that the manuscript, images, and Supporting Information files will be freely available online, and any third party is permitted to access, download, copy, distribute, and use these materials in any way, even commercially, with proper attribution. For more information, see our copyright guidelines: http://journals.plos.org/plosone/s/licenses-and-copyright.

a) You may seek permission from the original copyright holder of Figure 1 to publish the content specifically under the CC BY 4.0 license. 

Reviewers' comments:

Reviewer's Responses to Questions

**Comments to the Author**

1. Is the manuscript technically sound, and do the data support the conclusions?

Reviewer #1: Partly

Reviewer #2: Partly

2. Has the statistical analysis been performed appropriately and rigorously? 

Reviewer #1: No

Reviewer #2: No

3. Have the authors made all data underlying the findings in their manuscript fully available?

Reviewer #1: No

Reviewer #2: No

4. Is the manuscript presented in an intelligible fashion and written in standard English?

Reviewer #1: Yes

Reviewer #2: Yes

5. Review Comments to the Author

Reviewer #1: The study investigated the capacity of patients with Parkinson’s disease to anticipate the object of sentences by measuring fixations in a visual-world paradigm. Patients with Parkinson’s disease did not show different patterns of fixation on the target (object) during sentence processing. There were differences between patients and controls for a small set of high motion verbs.

I am not surprised by the results. The analysis focused on the onset and increase of target fixation over time. However, the target was already cued by the agent (first noun) and later constrained by the verb. So, fixations on the target could be due in large part to its semantic relationship with the agent, and this aspect of language processing is well preserved in PD. This issue is not corrected by eliminating trials in which participants were already looking at the target at the beginning of the trial. Although I appreciate the authors’ rationale, it would be preferable to report the results with all trials, at least in supplementary materials. Authors should report analyses comparing the time taken by participants to stop looking at the agent distractor once the verb was presented. This word is still compatible with the agent, but disengagement from the agent distractor would be expected if people predicted the next word based on verb information. Focusing the analysis on the agent distractor and/or on a proportion between agent distractor and target fixations could be more informative. Authors might find this paper interesting:

Hochstadt, J. (2009). Set-shifting and the on-line processing of relative clauses in Parkinson's disease: Results from a novel eye-tracking method. Cortex, 45(8), 991-1011.

I provide detailed comments below.

Introduction

less commonly, outright dementia = dementia is not common at disease onset, but it is common when patients are followed longitudinally

Hely, M. A., Reid, W. G., Adena, M. A., Halliday, G. M., & Morris, J. G. (2008). The Sydney multicenter study of Parkinson's disease: the inevitability of dementia at 20 years. Movement disorders, 23(6), 837-844.

Cardona et al. proposed that action-language networks involve not only motor cortex and respective mirror neuron systems but also cortical-subcortical systems. = Expand the review of papers showing the involvement of mirror neurons and motor areas for verbs and action language processing.

If event knowledge deficits are a symptom of Parkinson’s disease, then cognitively intact participants with Parkinson’s disease may show impaired processing of verbs and of their event-based semantic associates, compared to healthy adults. Online language processing may be particularly challenging for people with PD, considering that healthy adults activate event knowledge both to process and to predict language as it unfolds in real time. = This hypothesis is interesting, but it is not clear how experimental manipulations can help distinguish between event knowledge, verb semantics (motion content, lexical aspect-telicity, etc.), thematic roles, etc. This should be clarified or acknowledged as a limitation.

Under Huettig’s account, deficits in the use of verb specific thematic fit information in PD would appear capable of disrupting the proper functioning of multiple language prediction mechanisms. = The argument made in this sentence appears somewhat circular. PD patients can’t use thematic information for language predictions because processing of thematic information is impaired. Rephrase or clarify.

Relatively few published studies have investigated predictive processing based on thematic fit in cognitively intact participants with PD. = The term “cognitively intact” is charged and full of implications. Speaking of participants who do not have self-reported and measurable signs of cognitive decline might be more appropriate (see Litvan et al. 2012 for discussion on the criteria for MCI in PD and lack of consensus on how they should be applied).

Thus, what remains unclear is whether language prediction deficits appear in PD in situations that require rapid integration of concepts for combinatorial processing, as in combining agent-based and verb-based sources of semantic information. = How to integrate the results of studies on lexical activation delay in PD into this question?

We hypothesized that because people with PD evince impairments in action and event semantic knowledge, they should show impaired online processing of sentences that require rapidly combining thematic fit information from an agent noun and a verb to predict the post-verb object (patient). = I don’t think the word “require” is appropriate in this context, since participants are not made aware of the real purpose of the task. They are never “required” to fixate the target as rapidly as possible (or at all).

Although attention, working memory, and executive function impairments have sometimes been implicated in language processing impairments in PD, the objective of the present study was to investigate combinatorial semantic language prediction abilities in a cohort without concomitant cognitive impairment. = Same issue as above. No reported and measurable signs of cognitive impairment would be preferable.

Aim 1 examined whether the inability to integrate multiple sources of thematic role

information, grounded in deficits in event knowledge, is a source of language impairment in PD. = Please provide more support and citations for 1) the idea that thematic roles are related to event knowledge and 2) the idea that event knowledge (script memory?) is impaired in PD

Methods

Norming studies for stimuli used in the predictive visual world paradigm task = It is difficult for readers to understand what the problem was and what justified the inclusion of additional stimuli. Please give an example of problematic items and replacements or move this paragraph further in the section.

Provide more information on normative scores for the telephone administrated MOCA. What is the cutoff score for normal cognition? Did all participants get a score above the cutoff?

Variability in item ratings correlate significantly with age (Pearson’s r = 0.03, p = 0.877) = The correlation is not significant

Three control participants and one PD participant scored > 1.5 but < 2 SD below normal limits on multiple tests and thus met criteria for mild cognitive impairment (Litvan et al., 2012). = Replace symbols by the words above and below, respectively. Also, the fact that those participants were not include further justifies not making strong claims about recruiting “cognitively intact” participants.

In 27.7% of predictive trials and 25.1% of non-predictive trials, participants were already fixating the target image just prior to sentence onset. Because the present study intended to examine only agent-, verb-, and target-driven increases in fixations on target objects, these target-anticipated trials were removed from the analysis. = Add a note explaining this in figure captions.

By the same token, why not eliminate sentences in which the participant was already fixating the agent distractor? No move is necessary until the verb.

Even so, by the end of the agent window, the proportion of fixations on the target interest area increased to approximately 36.1%, significantly higher than chance (χ2(1564) = 102.88, p < .0001) = By the end of the agent, two words were more likely to be fixated: the target and the agent distractor. I am not convinced that it is appropriate to run analyses that use 25% for each picture as the threshold for chance fixation at that point of the experiment.

Using the same colour coding for Figure 4 as for the other Figures is misleading because the readers needs to read the methods very carefully and look at the Appendix to realise that those words were not the same as those used in the main task.

Norming = was the agent/target relationship as strong as the agent/agent distractor relationship? Was that tested? What about the verb/target and verb/verb distractor relationship?

Rating of high vs. low motion verbs seems is based on a very small number of participants. This should be acknowledged as a limitation.

Conclusion

Thus, the fact that all of the sentences that participants heard in the present study were

semantically plausible is a strength of the study. = Some of the sentences included words that fit better in a non-count use (e.g., The shopper saves money vs. The shopper saves the money).

In addition, many of the distractor objects in the current study fit the sentences reasonably well, albeit at a lower probability than the target object. = The assumption that “chance” fixation is 25% at all points is not met. Analyses should not rely on chance level, or chance level should be specified clearly.

One limitation of the present study is that we did not carefully control the action content

of the verbs used in the predictive sentences. = This is an important limitation. There are other limitations related with words, such as the use of verbs that are homonyms with nouns (e.g., rocks), lack of control for frequency, and the use of written words in norming tasks vs. pictures in experiments. Pictures were not necessarily the most prototypical representations of the concept, and some pictures illustrated scenes/persons instead of simple objects (e.g., jungle, fugitive, courtroom).

Reviewer #2: A visual world study investigated predictive sentence processing in Parkinson’s disease. Participants heard sentences like “The fisherman rocks…” while viewing visual arrays with targets like a boat. In contrast to the hypotheses, predictive fixations to targets did not differ between participants with PD and controls.

This research has many strengths. Prediction has important theoretical implications and is of clear interest in the sentence processing literature. This study addresses novel and interesting questions about prediction in PD, which are likely to make an important addition to the literature. In addition, the introduction is clear and synthesises the two literatures well, and the method makes effective use of norming and controls. Before I am able to recommend publication, I encourage addressing the following weaknesses.

PREDICTION

The analyses do not compellingly address a fundamental issue: were participants predicting? The analyses establish that fixations to targets increased over time when participants heard predictive sentences (e.g., including prior to target word onset). However, a not dissimilar pattern was also observed in baseline (i.e., nonpredictive) sentences (e.g., see Figure 4, in which fixations to targets also increased prior to target word onset). It's also not clear how the targets compared to the various nontargets. Thus, I encourage reporting analyses that explicitly address whether the predictable targets were fixated significantly more than the nonpredictable nontargets, as is typical in the literature.

TARGET-ANTICIPATED TRIALS

Relatedly, I'm concerned that eliminating target-anticipated trials may create more problems than it solves. Among them:

(1) It makes it problematic to compare predictable targets and nonpredictable nontargets (and by extension, to address whether participants were predicting).

(2) It makes it problematic to compare targets and nontargets prior to sentence onset (and by extension, to asses potential extraneous biases).

(3) Among non-target-anticipated trials, participants may be more likely to fixate targets later in the trial because they weren’t exploring these visual stimuli earlier in the trial.

(4) Only 27% of sentences involved high motion; eliminating a further 25% likely adds further noise.

Thus, I encourage an alternative approach.

HIGH MOTION

Relatedly, I'm concerned that the high motion analysis may simply be too noisy to yield clear conclusions. The bottom-left plot of Figure 3 (i.e., controls + high motion) suggests that the verb-related and unrelated distractors diverged BEFORE verb onset (i.e., before they had relevant information), which is perhaps best explained by noise.

BATTERY

Participants with PD did not differ from controls on any of the measures in the neuropsychological battery (Table 1). Against this backdrop, the observed (i.e., visual world) similarities may be less surprising (i.e., to the extent that the visual world reflects another cognitive task). I wonder if this pattern is typical in the PD literature? In addition, it may be worthwhile to address prior individual differences research. For example, individual differences in memory, speed of processing, etc. have been linked to predictive sentence processing; if participants with PD did not differ from controls on these measures, then they might not be expected to differ in prediction:

Huettig, F., & Janse, E. (2016). Individual differences in working memory and processing speed predict anticipatory spoken language processing in the visual world. Language, Cognition and Neuroscience, 31(1), 80-93.

Kukona, A., Braze, D., Johns, C. L., Mencl, W. E., Van Dyke, J. A., Magnuson, J. S., ... & Tabor, W. (2016). The real-time prediction and inhibition of linguistic outcomes: Effects of language and literacy skill. Acta Psychologica, 171, 72-84.

METHOD AND ANALYSES

Finally, I encourage clarifying a handful of issues:

p12 – “Norming Study 1… revealed at least one problematic component in each of the original proposed stimuli sets...”; it’s not clear why some of these materials were used if problematic.

p17 – “These analyses were conducted using the cor.test() function in R version 3.5.1”; here and throughout, the discussion can be streamlined (e.g., captured by the analysis code).

p25 – “removal also helped ensure that participants’ eye movements reflected naïve predictions based on the words in the sentences rather than predictions based on the study structure”; this raises the question, was there such structure? For example, did repeating the visual stimuli allow participants to predict targets independent of the sentences (e.g., such that once the boat was a target, it was never again a target on its subsequent presentations).

p25 – “All intermediate proportions of fixations on each interested area were rounded to either 0 (no fixation) or 1 (fixation) because the raw binned proportions essentially followed a binomial distribution”; it’s not clear why the raw eye movement data was binned if a binomial approach was to be used (i.e., prior to binning, it was presumably binomial).

p26 – “Linear term estimates indicate whether fixation proportions increase, decrease, or remain flat…”; the (i.e., logistic) analyses are modelling transformed binomial outcomes, not the curves depicted in the figures. Thus, it’s not clear if curvilinear forms are suitable.

p26 – “In each model, we additionally generated random intercepts and random linear slopes for subjects (to assess individual differences) and for items (to assess stimulus-driven variability)”; growth curve analysis is widely used to model participant fixation curves that are generated by averaging across visual world trials (i.e., yielding curvilinear forms). In contrast, it’s not clear if curvilinear forms are suitable for modelling trial-level binomial outcomes.

p28 – “by the end of the agent window, the proportion of fixations on the target interest area increased to approximately 36.1%, significantly higher than chance”; I encourage addressing important nuances within this time course. For example, was the boat fixated signifiantly more than the cradle and quilt but the same as the net?

6. PLOS authors have the option to publish the peer review history of their article (what does this mean?). If published, this will include your full peer review and any attached files.

Reviewer #1: No

Reviewer #2: No

---

## [Author Response · Author response to Decision Letter 0]

8 Oct 2021

Daniel Mirman, PhD

Academic Editor

PLOS ONE 

Dear Dr. Mirman, 

Thank you for inviting us to submit a revised draft of our manuscript entitled "Predictive language comprehension in Parkinson’s disease", PONE-D-21-22350, to PLOS ONE. We also appreciate the time and effort you and each of the reviewers have dedicated to providing insightful feedback on ways to strengthen our paper. Thus, it is with great pleasure that we resubmit our article for further consideration. We have incorporated changes that reflect the detailed suggestions you have graciously provided. We also hope that our edits and the responses we provide below satisfactorily address all the issues and concerns you and the reviewers have noted.

To facilitate your review of our revisions, the following is a point-by-point response to the questions and comments delivered in your letter dated 8/2/2021. Page and line numbers refer to the ‘Manuscript’ version without visible tracked changes.

EDITOR SUGGESTIONS:

• RESPONSE: We have updated the manuscript formatting to adhere to PLOS ONE’s style requirements.

• RESPONSE: We have removed the funding statements from the manuscript and ensured the accuracy of the funding information and financial disclosure sections of the online editor. Grant numbers are not created for Parkinson Canada (PSC) grants. We updated the funding to reflect NIH funding that was part of the reported work. While the specific grant details for the NIH funding have not been added to the manuscript, the obligatory NIH text that is required to appear in acknowledgment has been added to the revised version, specifically the proportion of the work that is attributed to NIH funding and the NIH disclosure statement.

• RESPONSE: We have updated our cover letter to reflect that experimental data and analysis scripts are now publicly available. The data availability statement within the manuscript has been updated with specific links and DOIs.

4. We note that Figure 1 in your submission contain copyrighted images. All PLOS content is published under the Creative Commons Attribution License (CC BY 4.0), …We require you to either (1) present written permission from the copyright holder to publish these figures specifically under the CC BY 4.0 license, or (2) remove the figures from your submission: a) You may seek permission from the original copyright holder of Figure 1 to publish the content specifically under the CC BY 4.0 license.

• RESPONSE: We have replaced the copyrighted images in Figure 1 with a representative figure for which we were able to obtain all permissions from the copyright holders.

REVIEWER 1 COMMENTS:

5. The study investigated the capacity of patients with Parkinson’s disease to anticipate the object of sentences by measuring fixations in a visual-world paradigm. Patients with Parkinson’s disease did not show different patterns of fixation on the target (object) during sentence processing. There were differences between patients and controls for a small set of high motion verbs.

I am not surprised by the results. The analysis focused on the onset and increase of target fixation over time. However, the target was already cued by the agent (first noun) and later constrained by the verb. So, fixations on the target could be due in large part to its semantic relationship with the agent, and this aspect of language processing is well preserved in PD. This issue is not corrected by eliminating trials in which participants were already looking at the target at the beginning of the trial. Although I appreciate the authors’ rationale, it would be preferable to report the results with all trials, at least in supplementary materials. Authors should report analyses comparing the time taken by participants to stop looking at the agent distractor once the verb was presented. This word is still compatible with the agent, but disengagement from the agent distractor would be expected if people predicted the next word based on verb information. Focusing the analysis on the agent distractor and/or on a proportion between agent distractor and target fixations could be more informative. Authors might find this paper interesting:

Hochstadt, J. (2009). Set-shifting and the on-line processing of relative clauses in Parkinson's disease: Results from a novel eye-tracking method. Cortex, 45(8), 991-1011.

• RESPONSE: We greatly appreciate this reviewer’s effort and their thoughtful and detailed comments. This reviewer is correct that, by design, there were only two likely options for the target object by the end of the verb window, although we believe that integration of verb information was required in order for participants to predominately fixate the target object over the agent distractor. We have performed additional analyses focused on fixations on the agent-related distractor and its differences from target fixations. Specifically, in the new section “Fixations on the agent-related distractor,” beginning on page 31, we have shown in an additional mixed effects model that proportions of fixation to the agent-related distractor decline significantly during the predictive window, reflecting disengagement from the agent distractor. In addition, a t-test comparing mean proportions of fixations on the target and agent distractor confirmed that participants fixate the target object significantly more often than the agent distractor during the predictive verb window. We have also re-analyzed fixations to the target, agent distractor, and verb distractor in predictive sentences with all trials included and have moved the original analyses from pages 30-36 to supplementary tables S9-S13.

6. less commonly, outright dementia = dementia is not common at disease onset, but it is common when patients are followed longitudinally

• RESPONSE: We agree that dementia, while less common at disease onset, is more common upon disease progression, and we have clarified our statement on p3, line 66.

7. Cardona et al. proposed that action-language networks involve not only motor cortex and respective mirror neuron systems but also cortical-subcortical systems. = Expand the review of papers showing the involvement of mirror neurons and motor areas for verbs and action language processing.

• RESPONSE: Thank you for your suggestion. We have expanded the review of this literature on pages 4-5.

8. If event knowledge deficits are a symptom of Parkinson’s disease, then cognitively intact participants with Parkinson’s disease may show impaired processing of verbs and of their event-based semantic associates, compared to healthy adults. Online language processing may be particularly challenging for people with PD, considering that healthy adults activate event knowledge both to process and to predict language as it unfolds in real time. = This hypothesis is interesting, but it is not clear how experimental manipulations can help distinguish between event knowledge, verb semantics (motion content, lexical aspect-telicity, etc.), thematic roles, etc. This should be clarified or acknowledged as a limitation.

• RESPONSE: We appreciate this reviewer’s insight. We have expanded the “Strengths and limitations” section on page 41 to incorporate this point as well as limitations identified in later comments.

9. Under Huettig’s account, deficits in the use of verb specific thematic fit information in PD would appear capable of disrupting the proper functioning of multiple language prediction mechanisms. = The argument made in this sentence appears somewhat circular. PD patients can’t use thematic information for language predictions because processing of thematic information is impaired. Rephrase or clarify.

• RESPONSE: We thank the reviewer for raising this issue. We have re-organized and partially re-written the section “Language prediction in PD” on pages 7-9 to clarify this point.

10. Relatively few published studies have investigated predictive processing based on thematic fit in cognitively intact participants with PD. = The term “cognitively intact” is charged and full of implications. Speaking of participants who do not have self-reported and measurable signs of cognitive decline might be more appropriate (see Litvan et al. 2012 for discussion on the criteria for MCI in PD and lack of consensus on how they should be applied).

• RESPONSE: We agree with this reviewer and have incorporated this suggestion throughout our paper.

11. Thus, what remains unclear is whether language prediction deficits appear in PD in situations that require rapid integration of concepts for combinatorial processing, as in combining agent-based and verb-based sources of semantic information. = How to integrate the results of studies on lexical activation delay in PD into this question?

• RESPONSE: This is an important question. We believe the inclusion of the baseline sentences helped us to assess potential differences between groups due to lexical activation delay. We have updated page 11, line 260 and page 12, line 272 to reflect this.

12. We hypothesized that because people with PD evince impairments in action and event semantic knowledge, they should show impaired online processing of sentences that require rapidly combining thematic fit information from an agent noun and a verb to predict the post-verb object (patient). = I don’t think the word “require” is appropriate in this context, since participants are not made aware of the real purpose of the task. They are never “required” to fixate the target as rapidly as possible (or at all).

• RESPONSE: Thank you for raising this point. We have rephrased this statement on page 9, lines 212-7. We think these changes now better communicate that we expect participants who integrate agent and verb information to predictively fixate the target, despite the fact that they are never “required” to do so. We hope that you agree.

13. Although attention, working memory, and executive function impairments have sometimes been implicated in language processing impairments in PD, the objective of the present study was to investigate combinatorial semantic language prediction abilities in a cohort without concomitant cognitive impairment. = Same issue as above. No reported and measurable signs of cognitive impairment would be preferable.

• RESPONSE: We agree; please see our response to critique #10 above.

14. Aim 1 examined whether the inability to integrate multiple sources of thematic role

information, grounded in deficits in event knowledge, is a source of language impairment in PD. = Please provide more support and citations for 1) the idea that thematic roles are related to event knowledge and 2) the idea that event knowledge (script memory?) is impaired in PD

• RESPONSE: We thank the reviewer for this comment. To make the link between thematic roles and event knowledge more transparent, we have raised this literature again before the presentation of the aims on page 9, lines 207-8. We have also expanded our rationale for hypothesizing event knowledge deficits in PD on page 5, lines 116-9.

15. Norming studies for stimuli used in the predictive visual world paradigm task = It is difficult for readers to understand what the problem was and what justified the inclusion of additional stimuli. Please give an example of problematic items and replacements or move this paragraph further in the section.

• RESPONSE: We regret that the initial description of the norming studies was unclear. In response to this comment, we have streamlined the explanation of the purpose of the norming studies on pages 12 and 13, lines 286-8, and we have expanded the description of problematic items on page 15, lines 337-9.

16. Provide more information on normative scores for the telephone administrated MOCA. What is the cutoff score for normal cognition? Did all participants get a score above the cutoff?

• RESPONSE: We have clarified the telephone-administered MoCA cutoff value which all participants scored above on page 13, lines 298-301.

17. Variability in item ratings correlate significantly with age (Pearson’s r = 0.03, p = 0.877) = The correlation is not significant

• RESPONSE: We thank the reviewer for this important correction. We have edited page 17, line 392 to reflect that this correlation was not significant.

18. Three control participants and one PD participant scored > 1.5 but < 2 SD below normal limits on multiple tests and thus met criteria for mild cognitive impairment (Litvan et al., 2012). = Replace symbols by the words above and below, respectively. Also, the fact that those participants were not include further justifies not making strong claims about recruiting “cognitively intact” participants.

• RESPONSE: We agree and have edited pages 19-20, lines 454-61 to reflect this comment.

19. In 27.7% of predictive trials and 25.1% of non-predictive trials, participants were already fixating the target image just prior to sentence onset. Because the present study intended to examine only agent-, verb-, and target-driven increases in fixations on target objects, these target-anticipated trials were removed from the analysis. = Add a note explaining this in figure captions. By the same token, why not eliminate sentences in which the participant was already fixating the agent distractor? No move is necessary until the verb.

• RESPONSE: See our response to critique #5; all trials are now included. We note also that our rationale for excluding target-anticipated trials was to exclude trials in which the participants may have predicted the target object due to images’ recurrence throughout the experiment; initial fixations to the agent distractor were not suggestive of prediction based on the study structure. 

20. Even so, by the end of the agent window, the proportion of fixations on the target interest area increased to approximately 36.1%, significantly higher than chance (χ2(1564) = 102.88, p < .0001) = By the end of the agent, two words were more likely to be fixated: the target and the agent distractor. I am not convinced that it is appropriate to run analyses that use 25% for each picture as the threshold for chance fixation at that point of the experiment.

• RESPONSE: With all trials analyzed, this analysis based on chance levels is no longer relevant to the discussion of findings during the agent window and has been deleted. We had selected 25% as the threshold because fixation proportions might be expected to rise to at least 25% even if no predictive information had been given during this time window, merely due to regression to the mean. We acknowledge that, due to the concurrent predictive information given in the sentence, actual fixation proportions should be higher than 25%.

21. Using the same colour coding for Figure 4 as for the other Figures is misleading because the readers needs to read the methods very carefully and look at the Appendix to realise that those words were not the same as those used in the main task.

• RESPONSE: We agree that re-using colors may cause reader confusion and have remade Figure 4 in greyscale. 

22. Norming = was the agent/target relationship as strong as the agent/agent distractor relationship? Was that tested? What about the verb/target and verb/verb distractor relationship?

• RESPONSE: We believe that the counterbalancing approach taken in the present study negates this issue, as items serve as their own distractor over the course of the study. We hope that the reviewer concurs and have expanded upon this point on page 14, lines 324-9, and on page 41-42, lines 867-73.

23. is based on a very small number of participants. This should be acknowledged as a limitation.

• RESPONSE: We concur; see our response to critique #8 in the limitations section.

24. Thus, the fact that all of the sentences that participants heard in the present study were

semantically plausible is a strength of the study. = Some of the sentences included words that fit better in a non-count use (e.g., The shopper saves money vs. The shopper saves the money).

• RESPONSE: We believe that this issue is also largely mitigated by the counterbalancing approach taken in the present study, as described in our response to critique #22. However, we have expanded our discussion of limitations on page 43 to address this concern.

25. In addition, many of the distractor objects in the current study fit the sentences reasonably well, albeit at a lower probability than the target object. = The assumption that “chance” fixation is 25% at all points is not met. Analyses should not rely on chance level, or chance level should be specified clearly.

• RESPONSE: See our response to critique #20.

26. One limitation of the present study is that we did not carefully control the action content

of the verbs used in the predictive sentences. = This is an important limitation. There are other limitations related with words, such as the use of verbs that are homonyms with nouns (e.g., rocks), lack of control for frequency, and the use of written words in norming tasks vs. pictures in experiments. Pictures were not necessarily the most prototypical representations of the concept, and some pictures illustrated scenes/persons instead of simple objects (e.g., jungle, fugitive, courtroom).

• RESPONSE: See our response to critique #8.

REVIEWER 2 COMMENTS:

27. The analyses do not compellingly address a fundamental issue: were participants predicting? The analyses establish that fixations to targets increased over time when participants heard predictive sentences (e.g., including prior to target word onset). However, a not dissimilar pattern was also observed in baseline (i.e., nonpredictive) sentences (e.g., see Figure 4, in which fixations to targets also increased prior to target word onset). It's also not clear how the targets compared to the various nontargets. Thus, I encourage reporting analyses that explicitly address whether the predictable targets were fixated significantly more than the nonpredictable nontargets, as is typical in the literature.

• RESPONSE: We greatly appreciate the time this reviewer took to provide detailed and thoughtful comments. In regards to this analysis, see our response to Reviewer 1, critique #5. Although we preferred to use mixed effects models in general due to their advantages in handling time series data, modeling polytomous categorical variables is a well-documented issue for mixed effects models (e.g., Barr, 2008). Therefore, we have performed an additional analysis that explicitly tests the difference in means between target and agent distractor fixations.

28. Relatedly, I'm concerned that eliminating target-anticipated trials may create more problems than it solves. Among them:

(1) It makes it problematic to compare predictable targets and nonpredictable nontargets (and by extension, to address whether participants were predicting).

(2) It makes it problematic to compare targets and nontargets prior to sentence onset (and by extension, to asses potential extraneous biases).

(3) Among non-target-anticipated trials, participants may be more likely to fixate targets later in the trial because they weren’t exploring these visual stimuli earlier in the trial.

(4) Only 27% of sentences involved high motion; eliminating a further 25% likely adds further noise.

• RESPONSE: See our response to critique #5 in response to Reviewer 1.

29. Relatedly, I'm concerned that the high motion analysis may simply be too noisy to yield clear conclusions. The bottom-left plot of Figure 3 (i.e., controls + high motion) suggests that the verb-related and unrelated distractors diverged BEFORE verb onset (i.e., before they had relevant information), which is perhaps best explained by noise.

• RESPONSE: We agree with the reviewer and have emphasized this point on page 39, line 824 and on page 42, lines 892-3.

30. Participants with PD did not differ from controls on any of the measures in the neuropsychological battery (Table 1). Against this backdrop, the observed (i.e., visual world) similarities may be less surprising (i.e., to the extent that the visual world reflects another cognitive task). I wonder if this pattern is typical in the PD literature? In addition, it may be worthwhile to address prior individual differences research. For example, individual differences in memory, speed of processing, etc. have been linked to predictive sentence processing; if participants with PD did not differ from controls on these measures, then they might not be expected to differ in prediction.

• RESPONSE: While this reviewer is correct about the influence of individual differences in visual world processing, our main objective was to investigate predictive processing abilities in PD based on combinatorial semantic processing, in the absence of cognitive impairment. We have clarified this point on page 10, lines 221-8.

31. p12 – “Norming Study 1… revealed at least one problematic component in each of the original proposed stimuli sets...”; it’s not clear why some of these materials were used if problematic.

• RESPONSE: We thank the reviewer for raising this point of confusion. We have revised pages 12-13, lines 286-8 and page 15, lines 340-2 to clarify that problematic stimuli were detected following the completion of norming study 1 and were replaced following the completion of norming study 2.

32. p17 – “These analyses were conducted using the cor.test() function in R version 3.5.1”; here and throughout, the discussion can be streamlined (e.g., captured by the analysis code).

• RESPONSE: We thank the reviewer for this suggestion. We have simplified the description of statistical analyses, except in the section “Analysis of eye-tracking data” where we felt that methodological detail was needed in the manuscript to allow the reader easily to evaluate the work.

33. p25 – “removal also helped ensure that participants’ eye movements reflected naïve predictions based on the words in the sentences rather than predictions based on the study structure”; this raises the question, was there such structure? For example, did repeating the visual stimuli allow participants to predict targets independent of the sentences (e.g., such that once the boat was a target, it was never again a target on its subsequent presentations).

• RESPONSE: The study structure did repeat the visual stimuli in the way this reviewer describes. We have clarified this rationale for removing these trials on page 31, lines 686-9.

34. p25 – “All intermediate proportions of fixations on each interested area were rounded to either 0 (no fixation) or 1 (fixation) because the raw binned proportions essentially followed a binomial distribution”; it’s not clear why the raw eye movement data was binned if a binomial approach was to be used (i.e., prior to binning, it was presumably binomial).

• RESPONSE: We have added a rationale for binning the raw data on page 27, lines 602-3.

35. p26 – “Linear term estimates indicate whether fixation proportions increase, decrease, or remain flat…”; the (i.e., logistic) analyses are modelling transformed binomial outcomes, not the curves depicted in the figures. Thus, it’s not clear if curvilinear forms are suitable. p26 – “In each model, we additionally generated random intercepts and random linear slopes for subjects (to assess individual differences) and for items (to assess stimulus-driven variability)”; growth curve analysis is widely used to model participant fixation curves that are generated by averaging across visual world trials (i.e., yielding curvilinear forms). In contrast, it’s not clear if curvilinear forms are suitable for modelling trial-level binomial outcomes.

• RESPONSE: This is an interesting query. We modelled trial-level data so that we were able to assess and control for random effects of trials due to the complex visual stimuli. We have consulted with statisticians specializing in mixed effects models to confirm that our analysis approach was appropriate and have now acknowledged these individuals. We have also cited relevant literature supporting this choice of statistical methods on page 29, lines 646-50. 

36. p28 – “by the end of the agent window, the proportion of fixations on the target interest area increased to approximately 36.1%, significantly higher than chance”; I encourage addressing important nuances within this time course. For example, was the boat fixated significantly more than the cradle and quilt but the same as the net?

• REPSONSE: See our response to Reviewer 1 #27. We chose to focus this supplementary analysis on a comparison between the target and the agent distractor, as our modeling approach limited comparisons between target and various distractor curves.

Again, thank you for giving us the opportunity to strengthen our manuscript with your valuable comments and queries. We have worked hard to incorporate your feedback and hope that these revisions persuade you to accept our submission.

---

## [Decision Letter · Decision Letter 1]

9 Nov 2021

PONE-D-21-22350R1Predictive language comprehension in Parkinson’s diseasePLOS ONE

Dear Dr. Roberts,

Thank you for submitting your revised manuscript to PLOS ONE. Both reviewers commend your response to their comments on your original submission. Reviewer 1 had no additional comments and Reviewer 2 had just a few minor suggestions for clarification and consistency. Therefore, we invite you to submit a revised version of the manuscript that addresses these points.

We look forward to receiving your revised manuscript.

Kind regards,

Daniel Mirman

Academic Editor

PLOS ONE

Journal Requirements:

Reviewers' comments:

Reviewer's Responses to Questions

**Comments to the Author**

1. If the authors have adequately addressed your comments raised in a previous round of review and you feel that this manuscript is now acceptable for publication, you may indicate that here to bypass the “Comments to the Author” section, enter your conflict of interest statement in the “Confidential to Editor” section, and submit your "Accept" recommendation.

Reviewer #1: All comments have been addressed

Reviewer #2: (No Response)

2. Is the manuscript technically sound, and do the data support the conclusions?

Reviewer #1: Yes

Reviewer #2: Yes

3. Has the statistical analysis been performed appropriately and rigorously? 

Reviewer #1: Yes

Reviewer #2: Yes

4. Have the authors made all data underlying the findings in their manuscript fully available?

Reviewer #1: Yes

Reviewer #2: Yes

5. Is the manuscript presented in an intelligible fashion and written in standard English?

Reviewer #1: Yes

Reviewer #2: Yes

6. Review Comments to the Author

Reviewer #1: All my comments have been addressed.

The system will not let me bypass the minimum character count, so I will add that I appreciate the efforts put into running additional analyses and integrating the comments in the revised version. The paper raises more questions than it answers, but suggestions for future studies are relevant and intriguing.

Reviewer #2: I thank the authors for their responses to my comments. The manuscript is strengthened throughout; this is particularly true of the results. I had just a few remaining minor suggestions:

1. The discussion is now clearer about the limitations of the motion results (e.g., “we interpret these results cautiously”; p39ln819). However, this isn’t reflected in the abstract, where these results receive considerable attention. Rather, I encourage acknowledging these limitations and/or limiting the discussion of these results in the abstract. Adding to this discussion (e.g., “it is unclear why a significant motion content x group interaction occurred before the onset of the verb”; p39ln821), I also encourage being explicit that these results are problematic because they imply that participants were sensitive to the content of the verbs before they’d heard them. (As a very minor comment, I think only one sentence type is described in the abstract before, “in either sentence type”; p2ln45.)

2. The verb-related distractor analyses are difficult to interpret without a baseline comparison. Paralleling the target vs. agent-related distractor t-test (p32ln710), I encourage including an analysis like a verb-related distractor vs. unrelated t-test. I also encourage reporting relevant descriptive statistics (M/SD). (As a very minor comment, I wonder if the Baseline analyses shouldn’t be presented before the Predictive analyses; as is, they almost seem unnecessary.)

3. Finally, I also encourage using graphs to capture the growth curve results in a clearer and more compelling way. For example, graphs depicting the measure under analysis (i.e., “our models predict the odds ratio of fixations on the target versus fixations on all other distractors. This odds ratio is log-transformed into “logits” of fixations on each interest area”; p28ln628) would be informative. Likewise, graphs depicting the growth curve fits would also be informative. Rather, it’s not clear how the intercept, linear, etc. growth curve results (e.g., see Table 2) map onto proportions of fixations (e.g., see Figure 2). (As a minor comment, the sample size was 48 participants, which seems at odds with, “when sample sizes are sufficiently large, e.g. >200 for linear estimates and >1000 for quadratic estimates, as was the case in the present study”; p29ln648.)

7. PLOS authors have the option to publish the peer review history of their article (what does this mean?). If published, this will include your full peer review and any attached files.

Reviewer #1: No

Reviewer #2: No

---

## [Author Response · Author response to Decision Letter 1]

24 Dec 2021

Thank you for inviting us to re-submit a revised draft of our manuscript entitled "Predictive language comprehension in Parkinson’s disease", PONE-D-21-22350, to PLOS ONE. We appreciate the continued support and feedback provided by you and each of the reviewers. Thus, it is with great pleasure that we resubmit our article for further consideration. We have incorporated changes that reflect the detailed suggestions you have graciously provided. We also hope that our edits and the responses we provide below satisfactorily address all the issues and concerns you and the reviewers have noted.

To facilitate your review of our revisions, the following is a point-by-point response to the questions and comments delivered in your letter dated 11/9/2021. Page and line numbers refer to the ‘Manuscript’ version without visible tracked changes.

EDITOR SUGGESTIONS:

• RESPONSE: We have adjusted referenced article titles to follow sentence capitalization consistently. We have also corrected the title in reference #44 (Copland et al. 2000).

REVIEWER 1 COMMENTS:

2. The system will not let me bypass the minimum character count, so I will add that I appreciate the efforts put into running additional analyses and integrating the comments in the revised version. The paper raises more questions than it answers, but suggestions for future studies are relevant and intriguing.

• RESPONSE: We would like to take this opportunity to thank this reviewer again for their comprehensive and insightful comments on the previous draft. 

REVIEWER 2 COMMENTS:

3. The discussion is now clearer about the limitations of the motion results (e.g., “we interpret these results cautiously”; p39ln819). However, this isn’t reflected in the abstract, where these results receive considerable attention. Rather, I encourage acknowledging these limitations and/or limiting the discussion of these results in the abstract. Adding to this discussion (e.g., “it is unclear why a significant motion content x group interaction occurred before the onset of the verb”; p39ln821), I also encourage being explicit that these results are problematic because they imply that participants were sensitive to the content of the verbs before they’d heard them. (As a very minor comment, I think only one sentence type is described in the abstract before, “in either sentence type”; p2ln45.).

• RESPONSE: We appreciate this reviewer’s concern and have revised the abstract (page 2, lines 52-4) and the discussion of the results (page 39, lines 834-6) accordingly. We have also revised page 2, line 45 to omit the implied reference to another sentence type.

4. The verb-related distractor analyses are difficult to interpret without a baseline comparison. Paralleling the target vs. agent-related distractor t-test (p32ln710), I encourage including an analysis like a verb-related distractor vs. unrelated t-test. I also encourage reporting relevant descriptive statistics (M/SD). (As a very minor comment, I wonder if the Baseline analyses shouldn’t be presented before the Predictive analyses; as is, they almost seem unnecessary.)

• RESPONSE: We have completed an additional t-test comparing looks to the verb-related and unrelated distractors (page 34, lines 735-9) and have updated page 32, lines 717-8 to include the requested descriptive statistics (M/SD). Although we appreciate this reviewer’s comment concerning the placement of the baseline sentences, we believe it best to present the predictive sentences first based on readers’ anticipated interest in the research questions addressed by these analyses and to follow with the baseline sentences in case of lingering questions about the reliability of the obtained eye-tracking results.

5. Finally, I also encourage using graphs to capture the growth curve results in a clearer and more compelling way. For example, graphs depicting the measure under analysis (i.e., “our models predict the odds ratio of fixations on the target versus fixations on all other distractors. This odds ratio is log-transformed into “logits” of fixations on each interest area”; p28ln628) would be informative. Likewise, graphs depicting the growth curve fits would also be informative. Rather, it’s not clear how the intercept, linear, etc. growth curve results (e.g., see Table 2) map onto proportions of fixations (e.g., see Figure 2). (As a minor comment, the sample size was 48 participants, which seems at odds with, “when sample sizes are sufficiently large, e.g. >200 for linear estimates and >1000 for quadratic estimates, as was the case in the present study”; p29ln648.)

• RESPONSE: We thank the reviewer for this suggestion and have created two new figures (Figs 3 and 6 in the current manuscript) that depict model fits versus data in logit units. We have also created two new supplemental files (S13 and S14 Figures) that depict Figures 2 and 5 in logit units, although we have chosen not to replace Figures 2 and 5 in the main body of the manuscript because we feel that many readers will find the proportion scale more familiar and intuitive for graphical interpretation. Per the reviewer’s comment about the apparent mismatch in sample size, we have revised page 29, lines 649-651 to clarify.

Again, thank you for giving us the opportunity to strengthen our manuscript with your valuable comments and queries. We have worked hard to incorporate your feedback and hope that these revisions persuade you to accept our submission.

---

## [Editor Report · Decision Letter 2]

27 Dec 2021

Predictive language comprehension in Parkinson’s disease

PONE-D-21-22350R2

Dear Dr. Roberts,

We’re pleased to inform you that your manuscript has been judged scientifically suitable for publication and will be formally accepted for publication once it meets all outstanding technical requirements. I'm sorry there were problems with the submission portal and I'm glad they were eventually resolved.

Kind regards,

Daniel Mirman

Academic Editor

PLOS ONE

---

## [Editor Report · Acceptance letter]

13 Jan 2023

PONE-D-21-22350R2 

Predictive language comprehension in Parkinson’s disease 

Dear Dr. Roberts:

I'm pleased to inform you that your manuscript has been deemed suitable for publication in PLOS ONE. Congratulations! Your manuscript is now with our production department. 

Kind regards, 

on behalf of

Dr. Daniel Mirman 

Academic Editor

PLOS ONE